# Deep Jump Learning for Off-Policy Evaluation in Continuous Treatment Settings

**Hengrui Cai** *
North Carolina State University
Raleigh, USA
hcai5@ncsu.edu

**Chengchun Shi** *
London School of Economics and Political Science
London, UK
C.Shi7@lse.ac.uk

**Rui Song**
North Carolina State University
Raleigh, USA
rsong@ncsu.edu

**Wenbin Lu**
North Carolina State University
Raleigh, USA
wlu4@ncsu.edu

## Abstract

We consider off-policy evaluation (OPE) in continuous treatment settings, such as personalized dose-finding. In OPE, one aims to estimate the mean outcome under a new treatment decision rule using historical data generated by a different decision rule. Most existing works on OPE focus on discrete treatment settings. To handle continuous treatments, we develop a novel estimation method for OPE using deep jump learning. The key ingredient of our method lies in adaptively discretizing the treatment space using deep discretization, by leveraging deep learning and multi-scale change point detection. This allows us to apply existing OPE methods in discrete treatments to handle continuous treatments. Our method is further justified by theoretical results, simulations, and a real application to Warfarin Dosing.

## 1 Introduction

Individualization proposes to leverage omni-channel data to meet individual needs. Individualized decision making plays a vital role in a wide variety of applications. Examples include individualized treatment regime in precision medicine [37; 3; 51; 5], customized pricing strategy in economics [38; 49], personalized recommendation system in marketing [32], etc. Prior to adopting any decision rule in practice, it is crucial to know the impact of implementing such a policy. In medical and public-policy domains, it is risky to apply a treatment decision rule or policy online to estimate its mean outcome [see, e.g., 34; 20; 29]. Off-policy evaluation (OPE) thus attracts a lot of attention by estimating the mean outcome under a new decision rule (or policy), i.e., the value function, using the offline data generated by a different decision rule.

Despite the popularity of developing OPE methods with finitely many treatment (or action) options [see e.g., 12; 11; 50; 54; 55; 30; 22; 48; 52; 13; 2; 53; 46; 23; 42; 43], less attention has been paid to the continuous treatment setting, such as personalized dose finding [4; 56; 57; 58], dynamic pricing [10], and contextual bandits [7]. Recently, a few OPE methods have been proposed to handle continuous treatments [25; 27; 45; 8; 44; 47; 24]. All these methods rely on the use of a kernel function to extend the inverse probability weighting (IPW) or doubly robust (DR) approaches developed in discrete treatment domains [see e.g., 12]. They suffer from two limitations. **First, the validity of these methods requires the mean outcome to be a smooth function over the treatment space**. This assumption could be violated in applications such as dynamic pricing, where the expected demand for a product has jump discontinuities as a function of the charged price [10].

---

*Equal contribution.

35th Conference on Neural Information Processing Systems (NeurIPS 2021).

Specifically, a product could attract a new segment of customers if the seller lowers the price below a certain threshold. Thus, there will be a sudden increase in demand by a small price reduction, yielding a discontinuous demand function. **Second, these kernel-based methods typically use a single bandwidth parameter**. This is sub-optimal in cases where the second-order derivative of the conditional mean function has an abrupt change in the treatment space; see Section 3.1 for details. Addressing these limitations requires the development of new policy evaluation tools and theory.

Our contributions are summarized as follows. Methodologically, we develop a deep jump learning (DJL) method by integrating deep learning [28], multi-scale change point detection [see e.g., 35, for an overview], and the doubly-robust value estimators in discrete domains. **Our method does not require kernel bandwidth selection. It does not suffer from the limitations of kernel-based methods.** The key ingredient of our method lies in adaptively discretizing the treatment space using deep discretization. This allows us to apply the IPW or DR methods to derive the value estimator. The discretization addresses the first limitation of kernel-based methods, allowing us to handle discontinuous value functions. The adaptivity addresses the second limitation of kernel-based methods. Specifically, it guarantees the optimality of the proposed method in cases where the second-order derivative of the conditional mean function has an abrupt change in the treatment space. Theoretically, we derive the convergence rate of the value estimator under DJL for any policy of interest, allowing the conditional mean outcome to be either a continuous or piecewise function of the treatment; see Theorems 1 and 2 for details. **Under the piecewise model assumption, the rate of convergence is faster than kernel-based OPE methods**. Under the continuous model assumption, kernel-based estimators converge at a slower rate when the bandwidth undersmoothes or oversmoothes the data. Proofs of our theorems rely on establishing the uniform rate of convergence of deep learning estimators; see Lemma E.1 in the supplementary article. We expect this result to be of general interest in contributing to the line of work on developing theories for deep learning methods [see e.g., 21; 39; 14]. Empirically, we show the proposed deep jump learning outperforms existing state-of-the-art OPE methods in both simulations and a real data application to warfarin dosing.

## 2 Preliminaries

We first formulate the OPE problem in continuous domains. We next review some related literature on the DR value estimator, kernel based evaluation methods, and multi-scale change point detection.

### 2.1 Problem Formulation

The observed offline datasets can be summarized into $\{(X_i, A_i, Y_i)\}_{1 \leq i \leq n}$ where $O_i = (X_i, A_i, Y_i)$ denotes the feature-treatment-outcome triplet for the $i$th individual and $n$ denotes the total sample size. We assume these data triplets are independent copies of the population variables $(X, A, Y)$. Let $\mathcal{X} \in \mathbb{R}^p$ and $\mathcal{A}$ denote the $p$ dimensional feature and treatment (or action) space, respectively. We focus on the setting where $\mathcal{A}$ is one-dimensional, as in personalized dose finding and dynamic pricing. A decision rule or policy $\pi : \mathcal{X} \to \mathcal{A}$ determines the treatment to be assigned given the observed feature. We use $b$ to denote the propensity score, also known as the behavior policy, that generates the observed data. Specifically, $b(\bullet|x)$ denotes the probability density function of $A$ given $X = x$. Define the expected outcome function conditional on the feature-treatment pair as

$$Q(x, a) = E(Y|X = x, A = a).$$

As standard in the OPE and the causal inference literature [see e.g., 4; 25], we assume the stable unit treatment value assumption, no unmeasured confounders assumption, and the positivity assumption are satisfied. The positivity assumption requires $b$ to be uniformly bounded away from zero. The latter two assumptions are automatically satisfied in randomized studies. These three assumptions guarantee that a policy's value is estimable from the observed data. Specifically, for a target policy $\pi$ of interest, its value can be represented by

$$V(\pi) = E[Q\{X, \pi(X)\}].$$

Our goal is to estimate the value $V(\pi)$ based on the observed data.

### 2.2 Doubly Robust Estimator and Kernel-Based Evaluation

For discrete treatments, Dudík et al. [12] proposed a DR estimator of $V(\pi)$ by

$$\frac{1}{n}\sum_{i=1}^{n} \psi(O_i, \pi, \widehat{Q}, \widehat{b}) = \frac{1}{n}\sum_{i=1}^{n}\left[\widehat{Q}\{X_i, \pi(X_i)\} + \frac{\mathbb{I}\{A_i = \pi(X_i)\}}{\widehat{b}(A_i|X_i)}\{Y_i - \widehat{Q}(X_i, A_i)\}\right], \quad (1)$$

where $\mathbb{I}(\bullet)$ denotes the indicator function, $\widehat{Q}$ and $\widehat{b}(a|x)$ denote some estimators for the conditional mean function $Q$ and the propensity score $b$, respectively. The second term inside the bracket corresponds to an augmentation term. Its expectation equals zero when $\widehat{Q} = Q$. The purpose of adding this term is to offer additional protection against potential model misspecification of $Q$. Such an estimator is doubly-robust in the sense that its consistency relies on either the estimation model of $Q$ or $b$ to be correctly specified. It can be semi-parametrically efficient whereas the importance sampling or direct method are not. By setting $\widehat{Q} = 0$, equation 1 reduces to the IPW estimator.

In continuous treatment domains, the indicator function $\mathbb{I}\{A_i = \pi(X_i)\}$ equals zero almost surely. Consequently, naively applying equation 1 yields a plug-in estimator $\sum_{i=1}^{n} \widehat{Q}\{X_i, \pi(X_i)\}/n$. To address this concern, the kernel-based methods proposed to replace the indicator function in equation 1 with a kernel function $K[\{A_i - \pi(X_i)\}/h]$, i.e.,

$$\frac{1}{n}\sum_{i=1}^{n} \psi_h(O_i, \pi, \widehat{Q}, \widehat{b}) = \frac{1}{n}\sum_{i=1}^{n}\left[\widehat{Q}\{X_i, \pi(X_i)\} + \frac{K\left\{\frac{A_i - \pi(X_i)}{h}\right\}}{\widehat{b}(A_i|X_i)}\{Y_i - \widehat{Q}(X_i, A_i)\}\right]. \qquad (2)$$

Here, the bandwidth $h$ represents a trade-off. The variance of the resulting value estimator decays with $h$. Yet, its bias increases with $h$. More specifically, it follows from Theorem 1 of [25] that the leading term of the bias is equal to

$$h^2 \frac{\int u^2 K(u) du}{2} E\left\{\left.\frac{\partial^2 Q(X, a)}{\partial a^2}\right|_{a=\pi(X)}\right\}. \qquad (3)$$

To ensure the term in equation 3 decays to zero as $h$ goes to 0, it requires the expected second derivative of the function $Q(x, a)$ exists, and thus $Q(x, a)$ needs to be a smooth function of $a$. However, as commented in the introduction, this assumption could be violated in certain applications.

### 2.3  Multi-Scale Change Point Detection

To adaptively discretize the treatment space, we leverage ideas from multi-scale change point detection literature. The change point analysis considers an ordered sequence of data, $Y_{1:n} = \{Y_1, \cdots, Y_n\}$, with unknown change point locations, $\tau = \{\tau_1, \cdots, \tau_K\}$ for some unknown integer $K$. Here, $\tau_i$ is an integer between 1 and $n - 1$ inclusive, and satisfies $\tau_i < \tau_j$ for $i < j$. These change points split the data into $K + 1$ segments. Assume there are sufficiently many data points lying within each segment such that the expected reward can be consistently estimated. Within each segment, the expected outcome is a constant function; see the left panel of Figure 1 for details. A number of methods have been proposed on estimating change points [see for example, 1; 26; 15; 17; 18, and the references therein], by minimizing a penalized objective function:

$$\arg\min_{\tau}\left(\frac{1}{n}\sum_{i=1}^{K+1}\left[\mathcal{C}\{Y_{(\tau_{i-1}+1):\tau_i}\}\right] + \gamma_n K\right),$$

where $\mathcal{C}$ is a cost function that measures the goodness-of-the-fit of the constant function within each segment and $\gamma_n K$ penalizes the number of change points. We remark that all the above cited works focused on either models without features or linear models. Our proposal goes beyond these works in that we consider models with features and use deep neural networks (DNN) to capture the complex relationship between the outcome and features.

## 3  Deep Jump Learning

In Section 3.1, we use a toy example to demonstrate the limitation of kernel-based methods. We present the main idea of our algorithm in Section 3.2. Details are given in Section 3.3. For simplicity, we set the action space $\mathcal{A} = [0, 1]$. Define a discretization $\mathcal{D}$ for the treatment space $\mathcal{A}$ as a set of mutually disjoint intervals $\{[\tau_0, \tau_1), [\tau_1, \tau_2), \ldots, [\tau_{K-1}, \tau_K]\}$ for some $0 = \tau_0 < \tau_1 < \tau_2 < \cdots < \tau_{K-1} < \tau_K = 1$ and some integer $K \geq 1$. The union of these intervals covers $\mathcal{A}$. We use $J(\mathcal{D})$ to denote the set of change point locations, i.e., $\{\tau_1, \cdots, \tau_{K-1}\}$. We use $|\mathcal{D}|$ to denote the number of intervals in $\mathcal{D}$ and $|\mathcal{I}|$ to denote the length of any interval $\mathcal{I}$.

### 3.1  Toy Example

As discussed in the introduction, existing kernel-based methods use a single bandwidth to construct the value estimator. Ideally, the bandwidth $h$ in the kernel $K[\{A_i - \pi(X_i)\}/h]$ shall vary with

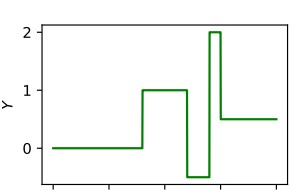 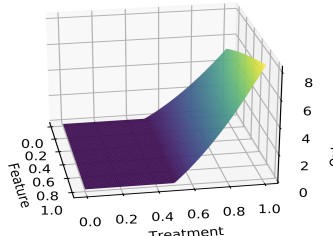 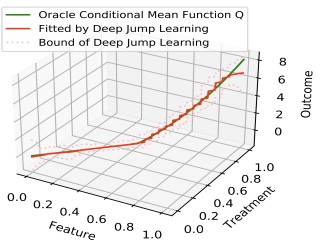

Figure 1: Left panel: example of piece-wise constant function. Middle panel: the oracle conditional mean function $Q$ on the feature-treatment space for the toy example. Right panel: the green curve presents the oracle $Q\{x, \pi(x)\}$ under target policy $\pi(x) = x$ in the toy example; and the red curve is the fitted mean value by DJL and the pink dash line corresponds to the 95% confidence bound.

Table 1: The absolute error and the standard deviation (in parentheses) of the estimated values for $V^{(1)}$ and $V^{(2)}$, using DJL and kernel-based methods, for target policy $\pi(x) = x$ in the toy example.

| Methods | Indicator | Deep Jump Learning | Kernel with $h = 0.4$ | Kernel with $h = 1$ |
|---------|-----------|--------------------|-----------------------|---------------------|
| $V^{(1)}(\pi)$ | $\mathbb{I}\{\pi(X) \leq 0.5\}$ | 0.31 (0.06) | 0.50 (0.08) | 0.40 (0.05) |
| $V^{(2)}(\pi)$ | $\mathbb{I}\{\pi(X) > 0.5\}$ | 0.09 (0.19) | 0.16 (0.20) | 1.09 (0.09) |

$\pi(X_i)$ to improve the accuracy of the value estimator. To elaborate this, consider the function $Q(x, a) = 10 \max(a^2 - 0.25, 0) \log(x + 2)$ for any $x, a \in [0, 1]$. By definition, $Q(x, a)$ is smooth over the entire feature-treatment space. However, it has different patterns when the treatment belongs to different intervals. Specifically, for $a \in [0, 0.5]$, $Q(x, a)$ is constant as a function of $a$. For $a \in (0.5, 1]$, $Q(x, a)$ depends quadratically in $a$. See the middle panel of Figure 1 for details.

Consider the target policy $\pi(x) = x$. We decompose the value $V(\pi)$ into $V^{(1)}(\pi) + V^{(2)}(\pi)$ where

$$V^{(1)}(\pi) = E[Q\{X, \pi(X)\}\mathbb{I}\{\pi(X) \leq 0.5\}], \text{ and } V^{(2)}(\pi) = E[Q\{X, \pi(X)\}\mathbb{I}\{\pi(X) > 0.5\}].$$

Similarly, denote the corresponding kernel-based value estimators by

$$\widehat{V}^{(1)}(\pi; h) = \frac{1}{n}\sum_{i=1}^{n}[\psi_h\mathbb{I}\{\pi(X_i) \leq 0.5\}], \widehat{V}^{(2)}(\pi; h) = \frac{1}{n}\sum_{i=1}^{n}[\psi_h\mathbb{I}\{\pi(X_i) > 0.5\}],$$

where $\psi_h := \psi_h(O_i, \pi, \widehat{Q}, \widehat{b})$ is defined in equation 2. Since $Q(x, a)$ is a constant function of $a \in [0, 0.5]$, its second-order derivative $\partial^2 Q(x, a)/\partial a^2$ equals zero. In view of equation 3, when $\pi(x) \leq 0.5$, the bias of $\widehat{V}^{(1)}(\pi; h)$ will be small even with a sufficiently large $h$. As such, a large $h$ is preferred to reduce the variance of $\widehat{V}^{(1)}(\pi; h)$. When $\pi(x) > 0.5$, a small $h$ is preferred to reduce the bias of $\widehat{V}^{(2)}(\pi; h)$. A simulation study is provided to demonstrate the drawback of the kernel-based methods. Specifically, we set $X, A \sim \text{Uniform}[0, 1]$ and generate $Y|X, A \sim N\{Q(X, A), 1\}$. We apply the kernel-based methods with a Gaussian kernel to estimate $V^{(1)}(\pi)$ and $V^{(2)}(\pi)$ with the sample size $n = 300$ over 100 replications. See Table 1 for details of the absolute error and standard deviation of $\widehat{V}^{(1)}(\pi; h)$ and $\widehat{V}^{(2)}(\pi; h)$ with two different bandwidths $h = 0.4$ and 1. It can be seen that **due to the use of a single bandwidth, the kernel-based estimator suffers from either a large absolute error or a large variance**.

To overcome this limitation, we propose to adaptively discretize the treatment space into a union of disjoint intervals such that within each interval $\mathcal{I}$, the conditional mean function $Q$ can be well-approximated by some functions $q_{\mathcal{I}}$ that depend on features but not on the treatment (constant in $a$), i.e., $Q(\bullet, a) \approx \sum_{\mathcal{I} \in \mathcal{D}}\{\mathbb{I}(a \in \mathcal{I})q_{\mathcal{I}}(\bullet)\}$. By the discretization, one can apply the IPW or DR methods to evaluate the value. The advantage of adaptive discretization is illustrated in the right panel of Figure 1, where we apply the proposed DJL method to the toy example. See details of the proposed method and its implementation in Sections 3.2 and 3.3. When $a \leq 0.5$, $Q(x, a)$ is constant in $a$. It is likely that our procedure will not further split the interval $[0, 0.5]$. Consequently, the corresponding DR estimator for $V^{(1)}(\pi)$ will not suffer from large variance. When $a > 0.5$, our procedure will split $(0.5, 1]$ into a series of sub-intervals, approximating $Q$ by a step function. This guarantees the resulting DR estimator for $V^{(2)}(\pi)$ will not suffer from large bias. Consequently, the proposed value estimator achieves a smaller mean squared error than kernel-based estimators. See Table 1 for details.

## 3.2 The Main Idea

We consider the following two model assumptions, which cover a variety of scenarios in practice.

**Model 1: Piecewise function.** Suppose

$$Q(x, a) = \sum_{\mathcal{I} \in \mathcal{D}_0} \left\{ q_{\mathcal{I},0}(x) \mathbb{I}(a \in \mathcal{I}) \right\}, \quad \text{for any } x \in \mathcal{X}, \text{ for any } a \in \mathcal{A}, \tag{4}$$

for some partition $\mathcal{D}_0$ of $[0, 1]$ and a collection of functions $\{q_{\mathcal{I},0}\}_{\mathcal{I} \in \mathcal{D}_0}$.

**Model 2: Continuous function.** Suppose $Q$ is a continuous function of $a$ and $x$.

Model 1 covers the dynamic pricing example mentioned in the introduction. In Section 5.1, the underlying model is set to be a piecewise function in Scenarios 1 and 2. Model 2 covers the personalized dose-finding example, Scenarios 3 and 4 in simulation studies, as well as the real data in Section 5.2. We next detail the proposed method, which will work when either Model 1 or 2 holds.

Motivated by Model 1, our goal is to identify an optimal discretization $\widehat{\mathcal{D}}$ such that for each interval $\mathcal{I} \in \widehat{\mathcal{D}}$, $Q(x, a)$ is approximately a constant function of $a \in \mathcal{I}$. Specifically, under Model 1, we assume the function $Q(x, a)$ is a piecewise function on the action space. Within each segment $\mathcal{I}$, the function $Q(x, a)$ is a constant function of $a$, but can be any function of the features $x$. In other words, $Q(x, a_1) = Q(x, a_2)$ for any $a_1, a_2 \in \mathcal{I}$. Thus, we denote the function $Q(x, a)$ at each segment $\mathcal{I}$ as $q_{\mathcal{I}}(x)$, which yields a piecewise function $Q(x, a) = \sum_{\mathcal{I}} q_{\mathcal{I}}(x) \mathbb{I}(a \in \mathcal{I})$, as stated in equation 4. In the real applications, the true function $Q(x, a)$ could be either a continuous function, or a piecewise function. As such, we propose to approximate the underlying unknown function $Q(x, a)$ by these piecewise functions of $a$ using DJL. Such approximation allows us to derive the DR estimator based on $\widehat{\mathcal{D}}$. The bias and variance of the resulting estimator are largely affected by the number of intervals in $\widehat{\mathcal{D}}$. Specifically, if $|\widehat{\mathcal{D}}|$ is too small, then the piecewise approximation is not accurate, leading to a biased estimator. If $|\widehat{\mathcal{D}}|$ is too large, then $\widehat{\mathcal{D}}$ will contain many short intervals, and the resulting estimator might suffer from a large variance.

To this end, we develop a data-adaptive method to compute $\widehat{\mathcal{D}}$. We first divide the treatment space $\mathcal{A}$ into $m$ disjoint intervals: $[0, 1/m), [1/m, 2/m), \ldots, [(m-1)/m, 1]$. We require the integer $m$ to diverge with the sample size $n$, such that the conditional mean function $Q$ can be well-approximated by a piecewise function on these intervals. Note that these $m$ initial intervals is not equal to $\widehat{\mathcal{D}}$, but only serve as the initial candidate intervals. Yet, $\widehat{\mathcal{D}}$ will be constructed by adaptively combining some of these intervals. We find in our numerical studies that the size of the final partition $|\widehat{\mathcal{D}}|$ is usually much less than $m$ (see Table 4 in Appendix C for more details). In practice, we recommend to set the initial number of intervals $m$ to be proportional to the sample size $n$, i.e., $m = n/c$ for some constant $c > 0$. The performance of the resulting value estimator is not overly sensitive to the choice of $c$.

We define $\mathcal{B}(m)$ as the set of discretizations $\mathcal{D}$ such that each interval $\mathcal{I} \in \mathcal{D}$ corresponds to a union of some of the $m$ initial intervals. Each discretization $\mathcal{D} \in \mathcal{B}(m)$ is associated with a set of functions $\{q_{\mathcal{I}}\}_{\mathcal{I} \in \mathcal{D}}$. We model these $q_{\mathcal{I}}$ using DNNs, to capture the complex dependence between the outcome and features. When $Q(\bullet, a)$ is well-approximated, we expect the least square loss $\sum_{\mathcal{I} \in \mathcal{D}} \sum_{i=1}^{n} [\mathbb{I}(A_i \in \mathcal{I}) \{Y_i - q_{\mathcal{I}}(X_i)\}^2]$, will be small. Thus, $\widehat{\mathcal{D}}$ can be estimated by solving

$$\left( \widehat{\mathcal{D}}, \{\widehat{q}_{\mathcal{I}} : \mathcal{I} \in \widehat{\mathcal{D}}\} \right) = \underset{\substack{\mathcal{D} \in \mathcal{B}(m), \\ \{q_{\mathcal{I}} \in \mathcal{Q}_{\mathcal{I}} : \mathcal{I} \in \mathcal{D}\}}}{\arg \min} \left( \sum_{\mathcal{I} \in \mathcal{D}} \left[ \frac{1}{n} \sum_{i=1}^{n} \mathbb{I}(A_i \in \mathcal{I}) \{Y_i - q_{\mathcal{I}}(X_i)\}^2 \right] + \gamma_n |\mathcal{D}| \right), \tag{5}$$

for some regularization parameter $\gamma_n$ and some function class of DNNs $\mathcal{Q}_{\mathcal{I}}$. Here, the penalty term $\gamma_n |\mathcal{D}|$ in equation 5 controls the total number of intervals in $\widehat{\mathcal{D}}$, as in multi-scale change point detection. A large $\gamma_n$ results in few intervals in $\widehat{\mathcal{D}}$ and a potential large bias of the value estimator, whereas a small $\gamma_n$ procedures a large number of intervals in $\widehat{\mathcal{D}}$, leading to a noisy value estimator. The theoretical order of $\gamma_n$ is detailed in Section 4. In practice, we use cross-validation to select $\gamma_n$. We refer to this step as deep discretization. Details of solving equation 5 are given in Section 3.3.

Given $\widehat{\mathcal{D}}$ and $\{\widehat{q}_{\mathcal{I}} : \mathcal{I} \in \widehat{\mathcal{D}}\}$, we apply the DR estimator in equation 1 to derive the value estimate for any target policy of interest $\pi$, i.e.,

$$\widehat{V}^{DR}(\pi) = \frac{1}{n} \sum_{\mathcal{I} \in \widehat{\mathcal{D}}} \sum_{i=1}^{n} \left( \mathbb{I}\{\pi(X_i) \in \mathcal{I}\} \left[ \frac{\mathbb{I}(A_i \in \mathcal{I})}{\widehat{b}_{\mathcal{I}}(X_i)} \{Y_i - \widehat{q}_{\mathcal{I}}(X_i)\} + \widehat{q}_{\mathcal{I}}(X_i) \right] \right), \tag{6}$$

where $\widehat{b}_{\mathcal{I}}(x)$ is some estimator of the generalized propensity score function $\text{pr}(A \in \mathcal{I}|X = x)$. We call this method as the deep jump learning.

### 3.3 The Complete Algorithm for Deep Jump Learning

We present the details for DJL in this section. To further reduce the bias of the value estimator in equation 6, we employ a data splitting and cross-fitting strategy [6]. That is, we use different subsets of data samples to estimate the discretization $\widehat{\mathcal{D}}$ and to construct the value estimator. Our algorithm consists of three steps: data splitting, deep discretization, and cross-fitting. We detail each step below.

**Step 1: data splitting.** We divide all $n$ samples into $\mathcal{L}$ disjoint subsets of equal size, where $\mathbb{L}_\ell$ denotes the indices of samples in the $\ell$th subset for $\ell = 1, \cdots, \mathcal{L}$. Let $\mathbb{L}_\ell^c = \{1, 2, \cdots, n\} - \mathbb{L}_\ell$ as the complement of $\mathbb{L}_\ell$. Data splitting allows us to use one part of the data, i.e., $\mathbb{L}_\ell^c$, to train models for the conditional mean and propensity score functions, and the remaining part, i.e., $\mathbb{L}_\ell$, to estimate the value. We aggregate the resulting estimates over different subsets to get full efficiency, as in Step 3.

**Step 2: deep discretization.** For each $\ell = 1, \cdots, \mathcal{L}$, we propose to apply deep discretization to compute a discretization $\widehat{\mathcal{D}}^{(\ell)}$ and $\{\widehat{q}_{\mathcal{I}}^{(\ell)} : \mathcal{I} \in \widehat{\mathcal{D}}^{(\ell)}\}$ by solving a version of equation 5 using the data subset in $\mathbb{L}_\ell^c$ only. We next present the computational details for solving this optimization. Our algorithm employs the pruned exact linear time method [26] to identify the change points with a cost function that involves DNN training. Specifically, for any interval $\mathcal{I}$, define $\widehat{q}_{\mathcal{I}}^{(\ell)}$ as the minimizer of

$$\underset{q_{\mathcal{I}} \in \mathcal{Q}_{\mathcal{I}}}{\arg\min} \frac{1}{|\mathbb{L}_\ell^c|} \sum_{i \in \mathbb{L}_\ell^c} \left[ \mathbb{I}(A_i \in \mathcal{I})\{q_{\mathcal{I}}(X_i) - Y_i\}^2 \right], \tag{7}$$

where $|\mathbb{L}_\ell^c|$ denotes the number of samples in $\mathbb{L}_\ell^c$. Define the cost function $\mathcal{C}^{(\ell)}(\mathcal{I})$ as the minimum value of the objective function equation 7, i.e, $\mathcal{C}^{(\ell)}(\mathcal{I}) = |\mathbb{L}_\ell^c|^{-1} \sum_{i \in \mathbb{L}_\ell^c} [\mathbb{I}(A_i \in \mathcal{I})\{\widehat{q}_{\mathcal{I}}^{(\ell)}(X_i) - Y_i\}^2]$. Computation of $\widehat{\mathcal{D}}^{(\ell)}$ relies on dynamic programming [16]. For any integer $1 \leq v^* < m$, denote by $\mathcal{B}(m, v^*)$ the set consisting of all possible discretizations $\mathcal{D}_{v^*}$ of $[0, v^*/m)$. Set $\mathcal{B}(m, m) = \mathcal{B}(m)$, we define the Bellman function as

$$\text{Bell}(v^*) = \inf_{\mathcal{D}_{v^*} \in \mathcal{B}(m, v^*)} \left\{ \sum_{\mathcal{I} \in \mathcal{D}_{v^*}} \mathcal{C}^{(\ell)}(\mathcal{I}) + \gamma_n(|\mathcal{D}_{v^*}| - 1) \right\}, \text{ and } \text{Bell}(0) = -\gamma_n.$$

Our algorithm recursively updates the Bellman cost function for $v^* = 1, 2, \cdots$ by

$$\text{Bell}(v^*) = \min_{v \in \mathcal{R}_{v^*}} \left\{ \text{Bell}(v) + \mathcal{C}^{(\ell)}([v/m, v^*/m)) + \gamma_n \right\}, \quad \text{for any } v^* \geq 1, \tag{8}$$

where $\mathcal{R}_{v^*}$ is the candidate change points list. For a given $v$, the right-hand-side of equation 8 corresponds to the cost of partitioning on a particular point. We then identify the best $v$ that minimizes the cost. This yields the Bellman function on $[0, v^*/m]$ on the left-hand-side. Here, the list of candidate change points $\mathcal{R}_{v^*}$ is given by

$$\left\{ v \in \mathcal{R}_{v^*-1} \cup \{v^* - 1\} : \text{Bell}(v) + \mathcal{C}^{(\ell)}([v/m, (v^* - 1)/m)) \leq \text{Bell}(v^* - 1) \right\}, \tag{9}$$

during each iteration with $\mathcal{R}_0 = \{0\}$. The constraint listed in equation 9 restricts the research space in equation 8 to a potentially much smaller set of candidate change points, i.e., $\mathcal{R}_{v*}$. The main purpose is to facilitate the computation by discarding change points not relevant to obtain the final discretization. It yields a linear computational cost [26]. In contrast, without these restrictions, it would yield a quadratic computational cost [16]. To solve equation 8, we search the optimal change point location $v$ that minimizes $\text{Bell}(v^*)$. This requires deep learning to estimate $\widehat{q}_{\mathcal{I}}^{(\ell)}$ and $\mathcal{C}^{(\ell)}(\mathcal{I})$ with $\mathcal{I} = [v/m, v^*/m)$ for each $v \in \mathcal{R}_{v^*}$. Let $v^1$ be the corresponding minimizer. We then define the change points list $\tau(v^*)$ as the set of change-point locations in $[0, v^*/m]$ computed by the dynamic programming algorithm. It is computed iteratively based on the update $\tau(v^*) = \{v^1, \tau(v^1)\}$, which means that during each iteration, it includes the current best change point location $v^1$ (that minimizes equation 8) and the previous change-point list for the interval $[0, v^1/m]$. This procedure is iterated to compute $\text{Bell}(v^*)$ and $\tau(v^*)$ for $v^* = 1, \ldots, m$, to find the best change-point set for interval $[0, 1]$. The optimal partition $\widehat{\mathcal{D}}^{(\ell)}$ is determined by the values stored in $\tau$. Specifically, we initialize $\widehat{\mathcal{D}}^{(\ell)} = [\tau(m)/m, 1]$, $r = m$ and recursively update $\widehat{\mathcal{D}}^{(\ell)}$ by setting $\widehat{\mathcal{D}}^{(\ell)} \leftarrow \widehat{\mathcal{D}}^{(\ell)} \cup [\tau(r)/m, r/m)$ and $r \leftarrow \tau(r)$, as in dynamic programming [16].

**Step 3: cross-fitting.** For each interval in the estimated optimal partition $\widehat{\mathcal{D}}^{(\ell)}$, let $\widehat{b}_{\mathcal{I}}^{(\ell)}(x)$ denote some estimator for the propensity score $\mathrm{pr}(A \in \mathcal{I}|X = x)$. In a randomized study, the density function $b(a|x)$ is known to us and we set $\widehat{b}_{\mathcal{I}}^{(\ell)}(x) = \int_{a \in \mathcal{I}} b(a|x)da$. To deal with data from observational studies, we estimate the generalized propensity score with deep learning using the training dataset $\mathbb{L}_\ell^c$ as $\widehat{b}_{\mathcal{I}}^{(\ell)}(x)$. We evaluate the target policy in each $\mathbb{L}_\ell$, based on $\widehat{q}_{\mathcal{I}}^{(\ell)}$, $\widehat{b}_{\mathcal{I}}^{(\ell)}$, and $\widehat{\mathcal{D}}^{(\ell)}$, trained in its complementary subsamples $\mathbb{L}_\ell^c = \{1, \cdots, n\} - \mathbb{L}_\ell$. Denote this value estimator for subset $\mathbb{L}_\ell$ as $\widehat{V}_\ell$. The final value estimator for $V(\pi)$ is to aggregate over $\widehat{V}_\ell$ for $\ell = 1, \cdots, \mathcal{L}$ via cross-fitting by

$$\widehat{V}(\pi) = \frac{1}{n}\sum_{\ell=1}^{\mathcal{L}}\sum_{\mathcal{I} \in \widehat{\mathcal{D}}^{(\ell)}}\sum_{i \in \mathbb{L}_\ell}\left[\mathbb{I}(A_i \in \mathcal{I})\frac{\mathbb{I}\{\pi(X_i) \in \mathcal{I}\}}{\widehat{b}_{\mathcal{I}}^{(\ell)}(X_i)}\{Y_i - \widehat{q}_{\mathcal{I}}^{(\ell)}(X_i)\} + \mathbb{I}(A_i \in \mathcal{I})\widehat{q}_{\mathcal{I}}^{(\ell)}(X_i)\right]. \quad (10)$$

Note the samples used to construct $\widehat{V}$ inside bracket are independent from those to estimate $\widehat{q}_{\mathcal{I}}^{(\ell)}$, $\widehat{b}_{\mathcal{I}}^{(\ell)}$ and $\widehat{\mathcal{D}}^{(\ell)}$. This helps remove the bias induced by overfitting in the estimation of $\widehat{q}_{\mathcal{I}}^{(\ell)}$, $\widehat{b}_{\mathcal{I}}^{(\ell)}$ and $\widehat{\mathcal{D}}^{(\ell)}$.

We give the detailed pseudocode in Algorithm 1 in Appendix B due to page limit. The computational complexity of the proposed approach is $\mathcal{O}(mC_n)$, where $C_n$ is the computational complexity of training one DNN model with the sample size $n$. Detailed analysis is provided in Appendix A. The code is publicly available at our repository at `https://github.com/HengruiCai/DJL`.

# 4 Theory

We investigate the theoretical properties of the proposed DJL method. All the proofs are provided in the supplementary article. Without loss of generality, assume the support $\mathcal{X} = [0, 1]^p$. To simplify the analysis, we focus on the case where the behavior policy $b$ is known to us, which automatically holds for data from randomized studies. We focus on the setting where the conditional mean function $Q$ is a smooth function of the features; see A1 below. Specifically, define the class of $\beta$-smooth functions, also known as Hölder smooth functions with exponent $\beta$, as

$$\Phi(\beta, c) = \left\{h : \sup_{\|\alpha\|_1 \le \lfloor\beta\rfloor}\sup_{x \in \mathcal{X}}|\Delta^\alpha h(x)| \le c, \sup_{\|\alpha\|_1 = \lfloor\beta\rfloor}\sup_{x,z \in \mathcal{X}, x \neq z}\frac{|\Delta^\alpha h(x) - \Delta^\alpha h(z)|}{\|x - z\|_2^{\beta - \lfloor\beta\rfloor}} \le c\right\},$$

for some constant $c > 0$, where $\lfloor\beta\rfloor$ denotes the largest integer that is smaller than $\beta$ and $\Delta^\alpha$ denotes the differential operator $\Delta^\alpha$ denote the differential operator: $\Delta^\alpha h(x) = \partial^{\|\alpha\|_1}h(x)/\partial x_1^{\alpha_1}\cdots\partial x_p^{\alpha_p}$, where $x = [x_1, \ldots, x_p]$. When $\beta$ is an integer, $\beta$-smoothness essentially requires a function to have bounded derivatives up to the $\beta$th order. The Hölder smoothness assumption is commonly imposed in the current literature [see e.g., 14]. Meanwhile, DJL is valid when $Q(x, a)$ is a nonsmooth function of $x$ as well [see e.g., 21]. Our theory thus can be further generalized to any function class that can be learned by neural nets at a certain rate. We introduce two conditions: (A1.) Suppose $b(a|\bullet) \in \Phi(\beta, c)$, and $Q(\bullet, a) \in \Phi(\beta, c)$ for any $a$; (A2.) Functions $\{\widehat{q}_{\mathcal{I}} : \mathcal{I} \in \widehat{\mathcal{D}}^{(\ell)}\}$ are uniformly bounded. Assumption (A2) ensures that the optimizer would not diverge in the uniform norm sense. Similar assumptions are commonly imposed in the literature to derive the convergence rates of DNN estimators [see e.g., 14]. Combining (A2) with (A1) allows us to derive the uniform rate of convergence for the class of DNN estimators $\{\widehat{q}_{\mathcal{I}} : \mathcal{I} \in \widehat{\mathcal{P}}\}$. Specifically, $\widehat{q}_{\mathcal{I}}$ converges at a rate of $O_p\{n|\mathcal{I}|^{-2\beta/(2\beta+p)}\}$ where the big-$O$ terms are uniform in $\mathcal{I}$, $p$ is the dimension of features. See Lemma E.1 in the supplementary article for details.

## 4.1 Properties under Model 1

We first consider Model 1 where the function $Q(x, a)$ takes the form of equation 4. As commented, this assumption holds in applications such as dynamic pricing. Without loss of generality, assume $q_{\mathcal{I}_1, 0} \neq q_{\mathcal{I}_2, 0}$ for any two adjacent intervals $\mathcal{I}_1, \mathcal{I}_2 \in \mathcal{D}_0$. This guarantees that the representation in equation 4 is unique. Let $L_{\mathcal{I}}$ and $W_{\mathcal{I}}$ be the number of hidden layers and total number of parameters in the function class of DNNs $\mathcal{Q}_{\mathcal{I}}$. Assume the number of change points in $\mathcal{D}_0$ is fixed. The following theorem summarizes the rate of convergence of the proposed estimator under Model 1.

**Theorem 1** *Suppose equation 4, (A1) and (A2) hold. Suppose $m$ is proportional to $n$, $Y$ is a bounded variable and $A$ has a bounded probability density function on $[0, 1]$. Assume $\{\gamma_n\}_{n \in \mathbb{N}}$ satisfies $\gamma_n \to 0$ and $\gamma_n \gg n^{-2\beta/(2\beta+p)}\log^8 n$. Then, there exist some classes of DNNs $\{\mathcal{Q}_{\mathcal{I}} : \mathcal{I}\}$ with $L_{\mathcal{I}} \asymp \log(n|\mathcal{I}|)$ and $W_{\mathcal{I}} \asymp (n|\mathcal{I}|)^{p/(2\beta+p)}\log(n|\mathcal{I}|)$ such that the following events occur with*

*probability at least $1 - O(n^{-2})$,*

*(i) $|\widehat{\mathcal{D}}^{(\ell)}| = |\mathcal{D}_0|$; and (ii) $\max_{\tau \in J(\mathcal{D}_0)} \min_{\widehat{\tau} \in J(\widehat{\mathcal{D}}^{(\ell)})} |\widehat{\tau} - \tau| = O\{n^{-2\beta/(2\beta+p)} \log^8 n\}$.*
*In addition, for any policy $\pi$ such that for any $\tau_0 \in J(\mathcal{D}_0)$, $pr\{\pi(X) \in [\tau_0 - \epsilon, \tau_0 + \epsilon]\} = O(\epsilon)$,*
*(iii) $\widehat{V}(\pi) = V(\pi) + O_p\{n^{-2\beta/(2\beta+p)} \log^8 n\} + O_p(n^{-1/2})$.*

We make a few remarks. First, the result in (i) imply that deep discretization correctly identifies the number of change points. The result in (ii) imply that any change point in $\mathcal{D}_0$ can be consistently identified. In particular, $J(\widehat{\mathcal{D}}^{(\ell)})$ corresponds to a subset of $\{1/m, 2/m, \cdots, (m-1)/m\}$. For any true change point $\tau$ in $\mathcal{D}_0$, there will be a change point in $\widehat{\mathcal{D}}^{(\ell)}$ that approaches $\tau$ at a rate of $n^{-2\beta/(2\beta+p)}$ up to some logarithmic factors. Second, it can be seen from the proof of Theorem 1 that the two error terms $O\{n^{-2\beta/(2\beta+p)} \log^8 n\}$ and $O(n^{-1/2})$ in (iii) correspond to the bias and standard deviation of the proposed value estimator, respectively. When $2\beta > p$, the bias term is negligible. A Wald-type confidence interval can be constructed to infer $V(\pi)$. The assumption $2\beta > p$ allow the deep learning estimator to converge at a rate faster than $n^{-1/4}$. Such a condition is commonly imposed in the literature for inferring the average treatment effect [see e.g., 6; 14]. When $\beta < p$, i.e., the underlying conditional mean function $Q$ is not smooth enough, the proposed estimator suffers from a large bias and might converge at a rate that is slower than the usual parametric rate. This concern can be addressed by employing the A-learning method [see e.g., 33; 40; 41]. The A-learning method is more robust and requires weaker conditions to achieve the parametric rate. Specifically, it only requires the difference $Q(x, 1) - Q(x, 0)$ to belongs to $\Phi(\beta, c)$. This is weaker than requiring both $Q(x, 1)$ and $Q(x, 0)$ to belongs to $\Phi(\beta, c)$. Third, to ensure the consistency of the proposed value estimator, we require that the distribution of the random variable $\pi(X)$ induced by the target policy does not have point-masses at the change point locations. This condition is mild. For nondynamic policies where $\pi(X) = \pi_0$ almost surely, it requires $\pi_0 \notin J(\mathcal{D}_0)$. We remark that the set $J(\mathcal{D}_0)$ has a zero Lebesgue measure on $[0, 1]$. For dynamic policies, it automatically holds when $\pi(X)$ has a bounded density on $[0, 1]$.

## 4.2 Properties under Model 2

We next consider Model 2 where the function $Q(x, a)$ is continuous in the treatment space.

**Theorem 2** *Assume $Q(x, a)$ is Lipschitz continuous, i.e., $|Q(x, a_1) - Q(x, a_2)| \leq L|a_1 - a_2|$, for all $a_1, a_2 \in [0, 1], x \in \mathcal{X}$, and some constant $L > 0$. Assume (A1) and (A2), and $m$ is proportional to $n$ and $\gamma_n$ is proportional to $\max\{n^{-3/5}, n^{-2\beta/(2\beta+p)} \log^9 n\}$. Then for any target policy $\pi$,*

$$\widehat{V}(\pi) - V(\pi) = O_p(n^{-1/5}) + O_p\{n^{-2\beta/(6\beta+3p)} \log^3 n\}.$$

When $4\beta > 3p$, the convergence rate is given by $O_p(n^{-1/5})$. We remark that the above upper bound is valid for any target policy $\pi$. The convergence rate in Theorem 2 may not be tight. To the best of our knowledge, no formal lower bounds of the value estimator have been established in the literature in the continuous treatment setting. In the literature on multi-scale change point detection, there are lower bounds on the estimated time series [see e.g., 1]. However, they considered settings without baseline covariates and it remains unclear how the rate of convergence of the estimated piecewise function can be translated into that of the value. We leave this for future research. Finally, we clarify our theoretical contributions compared with the deep learning theory established in Farrell et al. [14]. First, Farrell et al. [14] considered a single DNN, whereas we established the uniform convergence rate of several DNN estimators, since our proposal requires to train multiple DNN models. Establishing the uniform rate of convergence poses some unique challenges in deriving the results of Theorems 1 and 2. We need to control the initial number of the intervals $m$ to be proportional to $n$ and the order of penalty term $\gamma_n$, so that uniform convergence rate can be established across all intervals. To address this difficulty, we derive the tail inequality to bound the rate of convergence of the DNN estimator and use the Bonferroni's correction to establish the uniform rate of convergence.

## 4.3 Comparison with Kernel-Based Methods

To simplify the analysis, we assume the kernel function is symmetric, the nuisance function estimators $\widehat{Q}$ and $\widehat{b}$ are set to their oracle values $Q$ and $b$, and that $4\beta > 3p$. Suppose Model 1 holds. In Appendix D, we show that the convergence rate of kernel-based methods is given by $O_p(n^{-1/3})$ with optimal bandwidth selection. In contrast, the proposed estimator converges at a faster rate of $O_p(n^{-1/2})$. Suppose Model 2 holds. In Appendix D, we show that the convergence rate of kernel-based methods

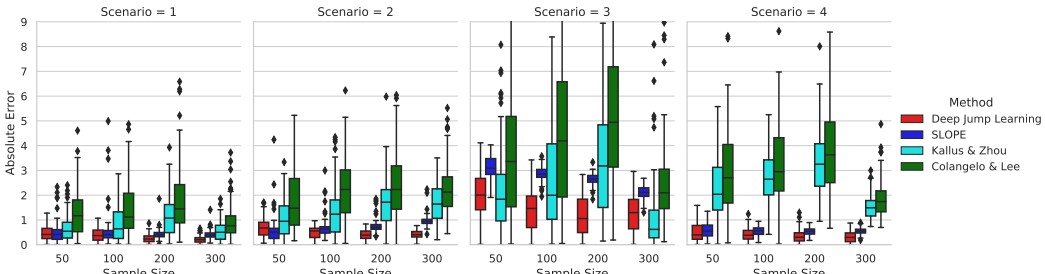

Figure 2: The box plot of the estimated values under the optimal policy via the proposed DJL and three kernel-based methods for Scenario 1-4. The target values are 1.33, 1, 4.86 and 1.6, respectively.

is given by $O_p(h) + O_p(n^{-1/2}h^{-1/2})$. Thus, kernel-based estimators converge at a slower rate when the bandwidth undersmoothes or oversmoothes the data. In addition, as we have commented in Section 3.1, in cases where the second-order derivative of $Q$ has an abrupt change in the treatment space, kernel-based methods suffer from either a large bias, or a large variance. Specifically, when $h$ is either much larger than $n^{-1/5}$ or much smaller than $n^{-3/5}$, our estimator converges at a faster rate. Kernel-based estimators could converge at a faster rate when $Q$ has a uniform degree of smoothness over the entire treatment space and the optimal bandwidth parameter is correctly identified.

## 5 Simulation Studies

In this section, we investigate the finite sample performance of the proposed method on the simulated and real datasets, in comparison to three kernel-based methods. The computing infrastructure used is a virtual machine in the AWS Platform with 72 processor cores and 144GB memory.

### 5.1 Simulation Settings

Simulated data are generated from the following model:
$Y|X, A \sim N\{Q(X, A), 1\}$, $b(A|X) \sim \text{Uniform}[0, 1]$ and $X^{(1)}, \dots, X^{(p)} \overset{iid}{\sim} \text{Uniform}[-1, 1]$,
where $X = [X^{(1)}, \cdots, X^{(p)}]$. Consider the following different scenarios:
**S1**: $Q(x, a) = (1 + x^{(1)})\mathbb{I}(a < 0.35) + (x^{(1)} - x^{(2)})\mathbb{I}(0.35 \leq a < 0.65) + (1 - x^{(2)})\mathbb{I}(a \geq 0.65)$;
**S2**: $Q(x, a) = \mathbb{I}(a < 0.25) + \sin(2\pi x^{(1)})\mathbb{I}(0.25 \leq a < 0.5) + \{0.5 - 8(x^{(1)} - 0.75)^2\}\mathbb{I}(0.5 \leq a < 0.75) + 0.5\mathbb{I}(a \geq 0.75)$;
**S3 (toy example)**: $Q(x, a) = 10 \max\{a^2 - 0.25, 0\} \log(x^{(1)} + 2)$;
**S4**: $Q(x, a) = 0.2(8 + 4x^{(1)} - 2x^{(2)} - 2x^{(3)}) - 2(1 + 0.5x^{(1)} + 0.5x^{(2)} - 2a)^2$.
The function $Q(x, a)$ is a piecewise function of $a$ under Scenarios 1 and 2, and is continuous under Scenarios 3 (toy example considered in Section 3.1) and 4. We set the target policy to be the optimal policy that achieves the highest possible mean outcome. The dimension of the features is fixed to $p = 20$. We consider four choices of the sample size, corresponding to $n = 50, 100, 200$ or $300$.

We compare the proposed DJL with three kernel-based methods [25; 8; 47]. In our implementation, we set $\mathcal{Q}_{\mathcal{I}}$ to the class of multilayer perceptrons (MLP) for each $\mathcal{I}$. This is a commonly used architecture in deep learning [14]. The optimization in equation 7 is solved via the MLP regressor implemented by Pedregosa et al. [36] using a stochastic gradient descent algorithm, with tuning parameters set to the default values. In addition, we estimate the propensity score function using MLP as well. We set $m = n/10$ to achieve a good balance between the absolute error and the computational cost (see Figure 1 in Appendix C for details). The averaged computational time are summarized in Table 1 with additional results under large sample sizes $n = 1000 \sim 10000$ in Table 2, in Appendix C. Overall, it takes a few minutes (less than 1 min for $n = 50$ and 14 mins for $n = 300$) to implement DJL, whereas the runtime of Kallus and Zhou [25]'s method is 365 mins for sample size $n = 50$ and over 48 hours for $n = 300$. Thus, as suggested in Kallus and Zhou [25], to implement their method, we first compute $h^*$ using data with sample size $n_0 = 50$. To accommodate data with different $n$, we adjust $h^*$ by setting $h^*\{n_0/n\}^{0.2}$. To implement Colangelo and Lee [8]'s estimator, we consider a list of bandwidths suggested in their paper, given by $h = c\sigma_A n^{-0.2}$ with $c \in \{0.5, 0.75, 1.0, 1.5\}$ and $\sigma_A$ is the standard deviation of the treatment. We then manually select the best bandwidth such that the resulting value estimator achieves the smallest mean squared error. The kernel-based method (SLOPE) by Su, Srinath and Krishnamurthy [47] adopted the Lepski's method for bandwidth selection. In their implementation, they used the IPW estimator to evaluate the value. For a fair comparison, we replace it with DR to make the resulting estimator more efficient.

Table 2: The absolute error, the standard deviation, and the mean squared error of the estimated values under the optimal policy via different methods for the Warfarin data. The target value is $-0.278$.

| Methods | Absolute error | Standard deviation | Mean squared error |
|---|---|---|---|
| Deep Jump Learning | 0.259 | 0.416 | 0.240 |
| SLOPE [47] | 0.611 | 0.755 | 0.943 |
| Kallus and Zhou [25] | 0.662 | 0.742 | 0.989 |
| Colangelo and Lee [8] | 0.442 | 1.164 | 1.550 |

The average estimated value and its standard deviation over 100 replicates are illustrated in Figure 2 for different methods, with detailed values reported in Table 3 in Appendix C. In addition, we provide the size of the final estimated partition under DJL in Table 4 in Appendix C, which is much smaller than $m$ in most cases. It can be seen from Figure 2 that DJL is very efficient and outperforms all competing methods in almost all cases. We note that the proposed method performs reasonably well even when the sample size is small ($n = 50$). In contrast, kernel-based methods fail to accurately estimate the value even in some cases when $n = 300$. Among the three kernel-based OPE approaches, we observe that the method developed by Su, Srinath and Krishnamurthy [47] performs better in general. A potential limitation of our method is that it takes a longer computational time than the method of Colangelo and Lee [8]. To speed up the dynamic programming algorithm, for instance, the total variation or group-fused-lasso-type penalty can be used as a surrogate of the $L_0$ penalty to reduce the computational complexity [see e.g., 19].

## 5.2    Real Data: Personalized Dose Finding

Warfarin is commonly used for preventing thrombosis and thromboembolism. We use the dataset provided by the International Warfarin Pharmacogenetics [9] for analysis. We choose $p = 81$ features considered in Kallus and Zhou [25]. This yields a total of 3964 with complete records. The outcome is defined as the absolute distance between the international normalized ratio (INR, a measurement of the time it takes for the blood to clot) after the treatment and the ideal value 2.5, i.e, $Y = -|\text{INR} - 2.5|$. We use the min-max normalization to convert the range of the dose level $A$ into $[0, 1]$. To compare among different methods, we calibrate the dataset to generate simulated outcomes. This allows us to use simulated data to calculate the bias and variance of each value estimator. Specifically, we first estimate the function $Q(x, a)$ via the MLP regressor using the whole dataset. The goodness-of-the-fit of the fitted model under the MLP regressor is reported in Table 5 in Appendix C. We next use the fitted function $\widehat{Q}(X, A)$ to simulate the data. For a given sample size $N$, we first randomly sample $N$ feature-treatment pairs $\{(a_j, x_j) : 1 \leq j \leq N\}$ from $\{(A_1, X_1), \cdots, (A_n, X_n)\}$ with replacement. Next, for each $j$, we generate the outcome $y_j$ according to $N\{\widehat{Q}(x_j, a_j), \widehat{\sigma}^2\}$, where $\widehat{\sigma}$ is the standard deviation of the fitted residual $\{Y_i - \widehat{Q}(X_i, A_i)\}_i$. This yields a simulated dataset $\{(x_j, a_j, y_j) : 1 \leq j \leq N\}$. We are interested in evaluating the mean outcome under the optimal policy as $\pi^\star(X) \equiv \arg\max_{a \in [0,1]} \widehat{Q}(X, a)$.

We apply DJL and the three kernel-based methods to the calibrated Warfarin dataset. Absolute errors, standard deviations, and mean squared errors of the estimated values under the optimal policy are reported in Table 2 over 20 replicates with different evaluation methods. It can be observed from Table 2 that DJL achieves much smaller absolute error (0.259) and standard deviation (0.416) than the three kernel-based methods. The mean square error of the three competing estimators are at least 3 times larger than DJL. The absolute error and standard deviation of Kallus and Zhou [25]'s estimator and of the SLOPE in Su, Srinath and Krishnamurthy [47] are approximately the same, due to that the bandwidth parameter is optimized. The estimator developed by Colangelo and Lee [8]'s performs the worst. It suffers from a large variance, due to the suboptimal choice of the bandwidth. All these observations are aligned with the findings in our simulation studies. Combining our theoretical analysis and experiments, we are more confident that DJL offers a practically much more useful policy evaluation tool compared to existing kernel-based approaches. There are some potential alternative directions to address the limitation of kernel-based approaches. Majzoubi et al. [31] proposed a tree-based discretization to handle continuous actions in policy optimization for contextual bandits. Extending the tree-based discretization with adaptive pruning in OPE is a possible direction to handle our problem. Second, DJL can be understood as a special local kernel method with the boxcar kernel function, as we adaptively discretize the action space into a set of non-overlapping intervals. It would be practically interesting to investigate how to couple our procedure with general kernel functions.

## Acknowledgments and Disclosure of Funding

The authors are grateful to the anonymous reviewers for valuable comments and suggestions. Chengchun Shi's research is partially supported by the London School of Economics and Political Science research support fund. Rui Song's research is partially supported by a grant from the National Science Foundation DMS-2113637.

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
