# Supplementary to 'Deep Jump Learning for Offline Policy Evaluation in Continuous Treatment Settings'

**Hengrui Cai** *
North Carolina State University
Raleigh, USA
hcai5@ncsu.edu

**Chengchun Shi** *
London School of Economics and Political Science
London, UK
C.Shi7@lse.ac.uk

**Rui Song**
North Carolina State University
Raleigh, USA
rsong@ncsu.edu

**Wenbin Lu**
North Carolina State University
Raleigh, USA
wlu4@ncsu.edu

## A  Analysis of Computational Complexity of DJL

We analyze the computational complexity for the proposed method as follows. There are three main dominating parts of the computation: the adaptive discretization, the estimations of conditional mean function and the propensity score function, and the construction of the final value estimator.

First, for the adaptive discretization on the treatment space (the main part of DJL, see Algorithm 1 Part III.3), we use the pruned exact linear time (PELT) method in Killick et al. (2012) to solve the dynamic programing. This step requires at least $\mathcal{O}(m)$ computing steps and at most $\mathcal{O}(m^2)$ steps (Friedrich et al. 2008). According to Theorem 3.2 in Killick et al. (2012), the expected computational cost is $\mathcal{O}(m)$.

Second, for each step in the linear complexity of adaptive discretization, we need to train the deep neural network for the conditional mean function and the propensity score function to calculate the cost function. Here, the time and space complexity of training a deep learning model varies depending on the actual architecture used. In our implementation, we employ the commonly used multilayer perceptron (MLP) to estimate the function $Q$ and the propensity score in each segment. Suppose we use the standard fully connected MLPs of $w$ width and $d$ depth with feedforward pass and back-propagation under total $e$ epochs. Then according to the complexity analysis of neural networks, the computational complexity of modeling the function $Q$ and the propensity score is $\mathcal{O}\{2 * ne(d-1)w^2\}$.

For the last part, the construction of the final value estimator based on $\mathcal{L}$-fold cross fitting, which repeats the above two steps $\mathcal{L}$ times. Therefore, by putting the above results together, the total expected computational complexity of the proposed DJL is $\mathcal{O}\{\mathcal{L} * m * 2 * ne(d-1)w^2\}$. Note that the computation for the last part (i.e., cross-fitting) can be easily implemented in parallel computing, and thus the total expected computational complexity of the proposed DJL can be reduced to $\mathcal{O}\{m * 2 * ne(d-1)w^2\}$.

## B  More on the implementation

We summarize our algorithm in Algorithm 1.

---

*Equal contribution.

35th Conference on Neural Information Processing Systems (NeurIPS 2021).

**Global:** data $\{(X_i, A_i, Y_i)\}_{1 \le i \le n}$; number of initial intervals $m$; penalty term $\gamma_n$; target policy $\pi$.
**Local:** Bellman function Bell $\in \mathbb{R}^m$; partitions $\widehat{\mathcal{D}}$; DNN functions $\{\widehat{q}_{\mathcal{I}}, \widehat{b}_{\mathcal{I}} : \mathcal{I} \in \widehat{\mathcal{D}}\}$; a vector $\tau \in \mathbb{N}^m$; a set of candidate point lists $\mathcal{R}$.
**Output:** the value estimator for target policy $\widehat{V}(\pi)$.

I. Split all $n$ samples into $\mathcal{L}$ subsets as $\{\mathbb{L}_1, \cdots, \mathbb{L}_{\mathcal{L}}\}$; $\widehat{V}(\pi) \leftarrow 0$;

II. Initialization:
    1. Set even segment on the action space with $m$ pieces:
       $\{\mathcal{I}\} = \{[0, 1/m), [1/m, 2/m), \ldots, [(m-1)/m, 1]\}$;
    2. Create a function to calculate cost $\mathcal{C}$ with inputs $(l, r)$:
       If $\mathcal{C}(l, r) == NULL$:
       (i). Let $\mathcal{I} = [l/m, r/m)$ if $r < m$ else $\mathcal{I} = [l/m, 1]$;
       (ii). Fit a DNN regressor: $\widehat{q}_{\mathcal{I}}(\cdot) \leftarrow \mathbb{I}(i \in \mathbb{L}_{\ell}^c)\mathbb{I}(A_i \in \mathcal{I})Y_i \sim \mathbb{I}(A_i \in \mathcal{I})DNN(X_i)$;
       (iii). Store the cost: $\mathcal{C}(\mathcal{I}) \leftarrow \sum_{i \in \mathbb{L}_{\ell}^c} \mathbb{I}(A_i \in \mathcal{I})\{\widehat{q}_{\mathcal{I}}(X_i) - Y_i\}^2$;
       Return $\mathcal{C}(l, r)$;

III. For $\ell = 1, \cdots, \mathcal{L}$:
    1. Set the training dataset as $\mathbb{L}_{\ell}^c = \{1, 2, \cdots, n\} - \mathbb{L}_{\ell}$;
    2. Bell$(0) \leftarrow -\gamma_n$; $\widehat{\mathcal{D}} = [0, 1]$; $\tau \leftarrow Null$; $\mathcal{R}(0) \leftarrow \{0\}$;
    3. Apply the pruned exact linear time method to get partitions: For $v^* = 1, \ldots, m$:
       (i).Bell$(v^*) = \min_{v \in \mathcal{R}(v^*)}\{$Bell$(v) + \mathcal{C}([v/m, v^*/m)) + \gamma_n\}$;
       (ii). $v^1 \leftarrow \arg\min_{v \in \mathcal{R}(v^*)}\{$Bell$(v) + \mathcal{C}([v/m, v^*/m)) + \gamma_n\}$;
       (iii). $\tau(v^*) \leftarrow \{v^1, \tau(v^1)\}$;
       (iv). $\mathcal{R}(v^*) \leftarrow \{v \in \mathcal{R}(v^*-1) \cup \{v^*-1\} : $Bell$(v) + \mathcal{C}([v/m, (v^*-1)/m)) \le $Bell$(v^*-1)\}$;
    4. Construct the DR value estimator: $r \leftarrow m$; $l \leftarrow \tau[r]$; While $r > 0$:
       (i) Let $\mathcal{I} = [l/m, r/m)$ if $r < m$ else $\mathcal{I} = [l/m, 1]$; $\widehat{\mathcal{D}} \leftarrow \widehat{\mathcal{D}} \cup \mathcal{I}$;
       (ii) Recall fitted DNN: $\widehat{q}_{\mathcal{I}}(\cdot) \leftarrow \mathbb{I}(i \in \mathbb{L}_{\ell}^c)\mathbb{I}(A_i \in \mathcal{I})Y_i \sim \mathbb{I}(A_i \in \mathcal{I})DNN(X_i)$;
       (iii) Fit propensity score: $\widehat{b}_{\mathcal{I}}(\cdot) \leftarrow \mathbb{I}(i \in \mathbb{L}_{\ell}^c)\mathbb{I}(A_i \in \mathcal{I}) \sim \mathbb{I}(A_i \in \mathcal{I})DNN(X_i)$;
       (iv) $r \leftarrow l$; $l \leftarrow \tau(r)$;
    6. Evaluation using testing dataset $\mathbb{L}_{\ell}$:
       $\widehat{V}(\pi)+ = \sum_{\mathcal{I} \in \widehat{\mathcal{D}}} \left(\sum_{i \in \mathbb{L}_{\ell}} \mathbb{I}(A_i \in \mathcal{I}) \left[\frac{\mathbb{I}\{\pi(X_i) \in \mathcal{I}\}}{\widehat{b}_{\mathcal{I}}(X_i)}\{Y_i - \widehat{q}_{\mathcal{I}}(X_i)\} + \widehat{q}_{\mathcal{I}}(X_i)\right]\right)$;
**return** $\widehat{V}(\pi)/n$ .

**Algorithm 1:** Deep Jump Learning

## C  Additional Experimental Results

We include additional experimental results in this section. First, the number of initial intervals $m$ represents a trade-off between the estimation bias and the computational cost, as illustrated in Figure 1. In practice, we recommend to set $m = n/10$. When $n$ is small, the performance of the resulting value estimator is not overly sensitive to the choice of $c$ as long as $c$ is not too large. See the left panel of Figure 1 for details. When $n$ is large, we further investigate the computational capacity of the proposed method by setting $m = n/10$ for large sample sizes and report the corresponding computational time in Table 2. We use Scenario 1 and consider the sample size chosen from $n \in \{1000, 2000, 5000, 10000\}$ for illustration. It turns out that such a choice of $c$ can still handle datasets with a few thousand observations. Here, we use parallel computing to process each fold, as our algorithm employs data splitting and cross-fitting. This largely facilitates the computation, leading to shorter computation time compared to those listed in Table 1. Finally, when $n$ is extremely large, setting $m = n/10$ might be computationally intensive. In addition to parallel computing, there are some other techniques we can use to handle datasets with large sample size. For instance, in the change-point literature, Lu et al. (2017) proposed an intelligence sampling method to identify multiple change points with long time series data. Their method would not lose much statistical efficiency, but is much more computationally efficient. It is possible to adopt such an intelligence sampling method to our setting for adaptive discretization. This would enable our method to handle large datasets.

Table 1: The averaged computational cost (in minutes) under the proposed deep jump learning and three kernel-based methods for Scenario 1.

| Methods | Deep Jump Learning | SLOPE (Su et al. 2020) | Kallus & Zhou (2018) | Colangelo & Lee (2020) |
|---|---|---|---|---|
| $n = 50$ | $< 1$ | <1 | 365 | $< 1$ |
| $n = 100$ | 3 | <1 | 773 | $< 1$ |
| $n = 200$ | 7 | 1 | $> 1440$ (24 hours) | $< 1$ |
| $n = 300$ | 14 | 2 | $> 2880$ (48 hours) | $< 1$ |

Table 2: The averaged computational cost under the proposed deep jump learning for Scenario 1 with large sample settings.

| Sample Size | $n = 1000$ | $n = 2000$ | $n = 5000$ | $n = 10000$ |
|---|---|---|---|---|
| Computational time | 15.92 minutes | 30.40 minutes | 1.32 hours | 2.86 hours |

Table 3: The absolute error and the standard deviation (in parentheses) of the estimated values under the optimal policy via the proposed deep jump learning and three kernel-based methods for Scenario 1 to 4.

| | $n$ | 50 | 100 | 200 | 300 |
|---|---|---|---|---|---|
| Scenario 1 | Deep Jump Learning | 0.445(0.381) | 0.398(0.391) | 0.253(0.269) | 0.209(0.210) |
| $V = 1.33$ | SLOPE (Su et al. 2020) | 0.392(0.377) | 0.385(0.549) | 0.329(0.400) | 0.344(0.209) |
| | Kallus & Zhou (2018) | 0.656(0.787) | 0.848(0.799) | 1.163(0.884) | 0.537(0.422) |
| | Colangelo & Lee (2020) | 1.285(1.230) | 1.473(1.304) | 1.826(1.463) | 0.934(0.730) |
| Scenario 2 | Deep Jump Learning | 0.696(0.376) | 0.502(0.311) | 0.400(0.219) | 0.411(0.168) |
| $V = 1.00$ | SLOPE (Su et al. 2020) | 0.620(0.634) | 0.859(0.822) | 0.749(0.878) | 1.209(0.435) |
| | Kallus & Zhou (2018) | 1.061(1.124) | 1.363(1.131) | 1.679(1.032) | 1.664(0.792) |
| | Colangelo & Lee (2020) | 1.827(1.371) | 2.292(1.458) | 2.429(1.541) | 2.264(1.062) |
| Scenario 3 | Deep Jump Learning | 2.014(0.865) | 1.410(0.987) | 1.184(0.967) | 1.267(0.933) |
| $V = 4.86$ | SLOPE (Su et al. 2020) | 3.660(0.496) | 3.185(0.592) | 2.897(0.781) | 2.037(0.401) |
| | Kallus & Zhou (2018) | 2.196(2.369) | 2.758(2.510) | 3.573(2.862) | 1.151(1.798) |
| | Colangelo & Lee (2020) | 2.586(2.825) | 3.172(3.027) | 3.949(3.391) | 1.367(2.110) |
| Scenario 4 | Deep Jump Learning | 0.494(0.485) | 0.412(0.426) | 0.349(0.383) | 0.321(0.315) |
| $V = 1.60$ | SLOPE (Su et al. 2020) | 0.586(0.337) | 0.537(0.279) | 0.483(0.272) | 0.483(0.143) |
| | Kallus & Zhou (2018) | 2.192(1.210) | 2.740(1.034) | 3.354(1.324) | 1.555(0.500) |
| | Colangelo & Lee (2020) | 2.975(1.789) | 3.282(1.525) | 3.921(1.927) | 1.853(0.751) |

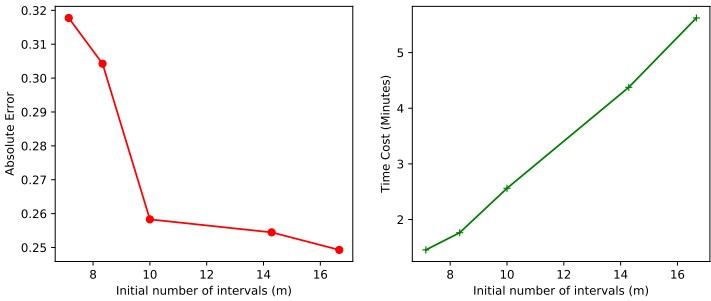

Figure 1: The absolute error of the estimated value and the computational cost (in minutes) under the DJL with different initial number of intervals ($m$) when $n = 100$ in Scenario 1.

Table 4: The averaged size of the final estimated partition ($|\widehat{\mathcal{D}}|$) in comparison to the initial number of intervals ($m$) under the proposed DJL for Scenario 1 to 4.

| $|\widehat{\mathcal{D}}|$ / $m$ | $n=50$ | $n=100$ | $n=200$ | $n=300$ |
|---|---|---|---|---|
| Scenario 1 | 3 / 5 | 4 / 10 | 6 / 20 | 6 / 30 |
| Scenario 2 | 4 / 5 | 6 / 10 | 9 / 20 | 11 / 30 |
| Scenario 3 | 4 / 5 | 6 / 10 | 8 / 20 | 10 / 30 |
| Scenario 4 | 4 / 5 | 6 / 10 | 8 / 20 | 10 / 30 |

Table 5: The mean squared error (MSE)[6], the normalized root-mean-square-deviation (NRMSD)[7], the mean absolute error (MAE)[8], and the normalized MAE (NMAE)[9] of the fitted model under the multilayer perceptrons regressor, linear regression, and the random forest algorithm, via ten-fold cross-validation.

| Method | Multilayer Perceptrons Regressor | Linear Regression | Random Forest |
|---|---|---|---|
| MSE | 0.06 | 0.09 | 0.08 |
| NRMSD | 0.13 | 0.16 | 0.15 |
| MAE | 0.19 | 0.23 | 0.22 |
| NMAE | 0.10 | 0.12 | 0.12 |

# D  Rate of Convergence of Kernel-Based Estimators

## D.1  Convergence Rate under Model 1

Consider the following piecewise constant function $Q$

$$Q(x,a) = \begin{cases} 0, & \text{if } a \leq 1/2, \\ 1, & \text{otherwise.} \end{cases}$$

Define a policy $\pi$ such that the density function of $\pi(X)$ equals

$$\begin{cases} 4/3, & \text{if } 1/4 \leq \pi(x) \leq 1/2, \\ 2/3, & \text{else if } 1/2 \leq \pi(x) < 4/3, \\ 0, & \text{otherwise.} \end{cases}$$

We aim to show for such $Q$ and $\pi$, the best possible convergence rate of kernel-based estimator is $n^{-1/3}$.

We first consider its variance. Suppose the conditional variance of $Y|A,X$ is uniformly bounded away from 0. Similar to Theorem 1 of Colangelo & Lee (2020), we can show the variance of kernel based estimator is lower bounded by $O(1)(nh)^{-1}$ where $O(1)$ denotes some positive constant.

We next consider its bias. Since the behavior policy is known, the bias is equal to

$$E\left(\frac{K[\{A-\pi(X)\}/h]}{hb(A|X)}[Y - Q\{X,\pi(X)\}]\right) = E\left(\frac{K[\{A-\pi(X)\}/h]}{hb(A|X)}[Q(X,A) - Q\{X,\pi(X)\}]\right)$$

$$= E\left(\int_{\pi(X)-h/2}^{\pi(X)+h/2} K\left\{\frac{a-\pi(X)}{h}\right\} [\mathbb{I}\{\pi(X) \leq 1/2 < a\} - \mathbb{I}\{a \leq 1/2 < \pi(X)\}]da\right).$$

Using the change of variable $a = ht + \pi(X)$, the bias equals

$$E\left(\int_{-1/2}^{1/2} K(t)[\mathbb{I}\{\pi(X) \leq 1/2 < \pi(X) + ht\} - \mathbb{I}\{\pi(X) + ht \leq 1/2 < \pi(X)\}]dt\right).$$

---

[6] $MSE = \frac{1}{n}\sum_{i=1}^{n}(Y_i - \widehat{Y}_i)^2$. See https://en.wikipedia.org/wiki/Mean_squared_error.
[7] $NRMSD = \frac{\sqrt{MSE}}{\max(Y)-\min(Y)}$. See https://en.wikipedia.org/wiki/Root-mean-square_deviation.
[8] $MAE = \frac{1}{n}\sum_{i=1}^{n}|Y_i - \widehat{Y}_i|$. See https://en.wikipedia.org/wiki/Mean_absolute_error.
[9] $NMAE = \frac{MAE}{\max(Y)-\min(Y)}$. See https://en.wikipedia.org/wiki/Root-mean-square_deviation.

Consider any $0 < h \le \epsilon$ for some sufficiently small $\epsilon > 0$. The bias is then equal to

$$\frac{4}{3} \int_{1/2-\epsilon/2}^{1/2} \int_{-1/2}^{1/2} K(t)\{\mathbb{I}(a \le 1/2 < a + ht) - \mathbb{I}(a + ht \le 1/2 < a)\}dtda$$

$$+ \frac{2}{3} \int_{1/2}^{1/2+\epsilon/2} \int_{-1/2}^{1/2} K(t)\{\mathbb{I}(a \le 1/2 < a + ht) - \mathbb{I}(a + ht \le 1/2 < a)\}dtda.$$

Under the symmetric condition on the kernel function, the above quantity is equal to

$$\frac{2}{3} \int_{1/2-h/2}^{1/2} \int_{(1-2a)/2h}^{1/2} K(t)dtda \ge \frac{2}{3} \int_{1/2-h/2}^{1/2-h/4} \int_{(1-2a)/2h}^{1/2} K(t)dtda$$

$$\ge \frac{2}{3} \int_{1/2-h/2}^{1/2-h/4} \int_{1/4}^{1/2} K(t)dtda = \frac{h}{6} \int_{1/4}^{1/2} K(t)dt.$$

Consequently, the bias is lower bounded by $O(1)h$ where $O(1)$ denotes some positive constant.

To summarize, the root mean squared error of kernel based estimator is lower bounded by $O(1)\{(nh)^{-1/2} + h\}$ where $O(1)$ denotes some positive constant. The optimal choice of $h$ that minimizes such lower bound would be of the order $n^{-1/3}$. Consequently, the convergence rate is lower bounded by $O(1)n^{-1/3}$.

## D.2 Convergence Rate under Model 2

Similar to the case under Model 1, we can show the variance of kernel-based estimator is lower bounded by $O(n^{-1}h^{-1})$ in cases where the conditional variance of $Y$ given $(A, X)$ is uniformly bounded away from zero.

Consider the conditional mean function $Q$

$$Q(x, a) = Ch^{-1}K\left\{\frac{a - \pi(x)}{h}\right\},$$

for some constant $C > 0$. We aim to derive the bias of kernel-based estimator under such a choice of the conditional mean function $Q$. Using similar arguments in the case where Model 1 holds, we can show the bias equals

$$E\left(C^{-1}\frac{K^2[\{A - \pi(X)\}/h]}{h^2 b(A|X)}\right) \ge C^{-1}E\left(\frac{K^2[\{A - \pi(X)\}/h]}{h^2}\right).$$

Similarly, we can show the right-hand-side is lower bounded by $O(1)h$. This implies that the convergence rate is at least $O(1)(n^{-1}h^{-1} + h)$ under Model 2.

# E  Technical Proof

Throughout the proof, we use $c, C, c_0, \bar{c}, c_*$, etc., to denote some universal constants whose values are allowed to change from place to place. Let $O_i = \{X_i, Y_i\}$ denote the data summarized from the $i$th observation. For any two positive sequences $\{a_n\}_n$ and $\{b_n\}_n$. The notation $a_n \asymp b_n$ means that there exists some universal constant $c > 1$ such that $c^{-1}b_n \le a_n \le cb_n$ for any $n$. The notation $a_n \propto b_n$ means that there exists some universal constant $c > 0$ such that $a_n \le cb_n$ for all $n$.

Proofs of Theorems 1 and 2 rely on Lemmas E.1, E.2 and E.3. In particular, Lemma E.1 establishes the uniform convergence rate of $\widehat{q}_{\mathcal{I}}^{(\ell)}$ for any $\mathcal{I}$ whose length is no shorter than $o(\gamma_n)$ and belongs to the set of intervals:

$$\mathfrak{I}(m) = \{[i_1/m, i_2/m) : \text{for some integers } i_1 \text{ and } i_2 \text{ that satisfy } 0 \le i_1 < i_2 < m\}$$
$$\cup \{[i_3/m, 1] : \text{for some integers } i_3 \text{ that satisfy } 0 \le i_3 < m\}.$$

To state this lemma, we first introduce some notations. For any such interval $\mathcal{I}$, define the function $q_{\mathcal{I},0}(x) = E(Y|A \in \mathcal{I}, X = x)$. It is immediate to see that the definition of $q_{\mathcal{I},0}$ here is consistent with the one defined in equation 4 for any $\mathcal{I} \subseteq \mathcal{D}_0$.

**Lemma E.1** *Assume either conditions in Theorem 1 or 2 are satisfied. Then there exists some constant $\bar{C} > 0$ such that the following holds with probability at least $1 - O(n^{-2})$: For any $1 \le \ell \le \mathcal{L}$, $\mathcal{I} \in \mathfrak{I}(m)$ and $|\mathcal{I}| \ge c\gamma_n$,*

$$E[|q_{\mathcal{I},0}(X) - \widehat{q}_{\mathcal{I}}^{(\ell)}(X)|^2 \{O_i\}_{i \in \mathbb{L}_\ell^c}] \le \bar{C}(n|\mathcal{I}|)^{-2\beta/(2\beta+p)} \log^8 n. \tag{1}$$

Here, the expectation in equation 1 is taken with respect to a testing sample $X$.

**Lemma E.2** *Assume either conditions in Theorem 1 or 2 are satisfied. Then there exists some constant $\bar{C} > 0$ such that the followings hold with probability at least $1 - O(n^{-2})$: For any $1 \le \ell \le \mathcal{L}$, $\mathcal{I} \in \mathfrak{I}(m)$ and $|\mathcal{I}| \ge c\gamma_n$,*

$$\sum_{\mathcal{I} \in \widehat{\mathcal{D}}^{(\ell)}} \left| \sum_{i \in \mathbb{L}_\ell^c} \mathbb{I}(A_i \in \mathcal{I})\{Y_i - q_{\mathcal{I},0}(X_i)\}\{\widehat{q}_{\mathcal{I}}^{(\ell)}(X_i) - q_{\mathcal{I},0}(X_i)\} \right| \le \bar{C}(n|\mathcal{I}|)^{p/(2\beta+p)} log^8 n.$$

**Lemma E.3** *Assume either conditions in Theorem 1 or 2 are satisfied. Then the following events occur with probability at least $1 - O(n^{-2})$: there exists some constant $c > 0$ such that $\min_{\mathcal{I} \in \widehat{\mathcal{D}}^{(\ell)}} |\mathcal{I}| \ge c\gamma_n$ for any $1 \le \ell \le \mathcal{L}$.*

We first present the proofs for these three lemmas. Next we present the proofs for Theorems 1 and 2.

### E.1 Proof of Lemma E.1

The number of folds $\mathcal{L}$ is bounded. It suffices to derive the uniform convergence rate for each $\ell$. By definition, $\widehat{q}_{\mathcal{I}}^{(\ell)}$ is the minimizer of the least square loss, $\arg\min_{q \in \mathcal{Q}_\mathcal{I}} \sum_{i \in \mathbb{L}_\ell^c} \mathbb{I}(A_i \in \mathcal{I})|Y_i - q(X_i)|^2$. It follows that

$$\sum_{i \in \mathbb{L}_\ell^c} \mathbb{I}(A_i \in \mathcal{I})|Y_i - \widehat{q}_{\mathcal{I}}^{(\ell)}(X_i)|^2 \le \sum_{i \in \mathbb{L}_\ell^c} \mathbb{I}(A_i \in \mathcal{I})|Y_i - q(X_i)|^2,$$

for all $q \in \mathcal{Q}_\mathcal{I}$. Recall that $q_{\mathcal{I},0}(x) = E(Y|A \in \mathcal{I}, X = x)$, we have $E[\mathbb{I}(A \in \mathcal{I})\{Y - q_{\mathcal{I},0}(X)\}|X] = 0$. A simple calculation yields

$$\sum_{i \in \mathbb{L}_\ell^c} \mathbb{I}(A_i \in \mathcal{I})|q_{\mathcal{I},0}(X_i) - \widehat{q}_{\mathcal{I}}^{(\ell)}(X_i)|^2 \le \sum_{i \in \mathbb{L}_\ell^c} \mathbb{I}(A_i \in \mathcal{I})|q_{\mathcal{I},0}(X_i) - q(X_i)|^2$$

$$+ 2\sum_{i \in \mathbb{L}_\ell^c} \mathbb{I}(A_i \in \mathcal{I})\{Y_i - q_{\mathcal{I},0}(X_i)\}\{\widehat{q}_{\mathcal{I}}^{(\ell)}(X_i) - q_{\mathcal{I},0}(X_i)\},$$

for any $q$ and $\mathcal{I}$.

The first term on the right-hand-side measures the approximation bias of the class of deep neural networks. Since $E[\mathbb{I}(A \in \mathcal{I})\{Y - q_{\mathcal{I},0}(X)\}|X] = 0$, the second term corresponds to the stochastic error. The rest of the proof is divided into three parts. In Part 1, we bound the approximation error. In Part 2, we bound the stochastic error. Finally, we combine these two parts together to derive the uniform convergence rate for $\widehat{q}_{\mathcal{I}}^{(\ell)}$.

*Part 1.* Under the given condition, we have $Q(\bullet, a) \in \Phi(\beta, c)$, $b(a|\bullet) \in \Phi(\beta, c)$ for some $c > 0$ and any $a$. We now argue that there exists some constant $C > 0$ such that $q_{\mathcal{I},0} \in \Phi(\beta, C)$ for any $\mathcal{I}$. This can be proven based on the relation that

$$q_{\mathcal{I},0}(x) = \frac{\int_\mathcal{I} Q(x,a)b(a|x)da}{\int_\mathcal{I} b(a|x)da}.$$

Specifically, we have that $\sup_x |q_{\mathcal{I},0}(x)| \le \sup_{a,x} |Q(x,a)| \le c$. Suppose $\beta \le 1$. For any $x_1, x_2 \in \mathcal{X}$, consider the difference $|q_{\mathcal{I},0}(x_1) - q_{\mathcal{I},0}(x_2)|$. Under the positivity assumption, we have $\inf_{a,x} b(a|x) \ge c_*$ for some $c_* > 0$. It follows that

$$|q_{\mathcal{I},0}(x_1) - q_{\mathcal{I},0}(x_2)| \le \frac{\int_\mathcal{I} |Q(x_1,a) - Q(x_2,a)|b(a|x_1)da}{\int_\mathcal{I} b(a|x_1)da}$$

$$+ \frac{\int_\mathcal{I} |Q(x_2,a)||b(a|x_1) - b(a|x_2)|da}{\int_\mathcal{I} b(a|x_1)da} + \frac{\int_\mathcal{I} |Q(x_2,a)|b(a|x_2)da \int_\mathcal{I} |b(a|x_1) - b(a|x_2)|da}{\int_\mathcal{I} b(a|x_1)da \int_\mathcal{I} b(a|x_2)da}$$

$$\le c\|x_1 - x_2\|^{\beta - \lfloor\beta\rfloor} + 2\frac{c^2}{c_*}\|x_1 - x_2\|^{\beta - \lfloor\beta\rfloor}.$$

Consequently, $q_{\mathcal{I},0} \in \Phi(\beta, c + 2c^2/c_*^2)$.

Suppose $\beta > 1$. Then both $Q(\bullet, a)$ and $b(a|\bullet)$ are $\lfloor\beta\rfloor$-differentiable. By changing the order of integration and differentiation, we can show that $q_{\mathcal{I},0}(x)$ is $\lfloor\beta\rfloor$-differentiable as well. As an illustration, when $\beta < 2$, we have $\lfloor\beta\rfloor = 1$. According to the chain rule, we have

$$\frac{\partial q_{\mathcal{I},0}(x)}{\partial x^j} = \frac{\int_{\mathcal{I}} \{\partial Q(x,a)/\partial x^j\} b(a|x) da}{\int_{\mathcal{I}} b(a|x) da} + \frac{\int_{\mathcal{I}} Q(a|x)\{\partial b(a|x)/\partial x^j\} da}{\int_{\mathcal{I}} b(a|x) da}$$
$$- \frac{\int_{\mathcal{I}} Q(a|x) b(a|x) da \int_{\mathcal{I}} \{\partial b(a|x)/\partial x^j\} da}{\{\int_{\mathcal{I}} b(a|x) da\}^2}.$$

Moreover, using similar arguments in proving $q_{\mathcal{I},0} \in \Phi(\beta, c + 2c^2/c_*^2)$ when $\beta < 1$, we can show that all the partial derivatives of $q_{\mathcal{I},0}(x)$ up to the $\lfloor\beta\rfloor$th order are uniformly bounded for all $\mathcal{I}$. In addition, all the $\lfloor\beta\rfloor$th order partial derivatives are Hölder continuous with exponent $\beta - \lfloor\beta\rfloor$. This implies that $q_{\mathcal{I},0} \in \Phi(\beta, C)$ for some constant $C > 0$ and any $\mathcal{I}$.

It is shown in Lemma 7 of Farrell et al. (2021) that for any $\epsilon > 0$, there exists a deep neural network architecture that approximates $q_{\mathcal{I},0}$ with the uniform approximation error upper bounded by $\epsilon$, and satisfies $W_{\mathcal{I}} \le \bar{C}\epsilon^{-p/\beta}(\log \epsilon^{-1} + 1)$ and $L_{\mathcal{I}} \le \bar{C}(\log \epsilon^{-1} + 1)$ for some constant $\bar{C} > 0$. These upper bounds will be used later in Part 2. The detailed value of $\epsilon$ will be specified below. It follows that for any $\mathcal{I}$, the bias term can be upper bounded by

$$\sum_{i \in \mathbb{L}_\ell^c} \mathbb{I}(A_i \in \mathcal{I})|q_{\mathcal{I},0}(X_i) - q(X_i)|^2 \le \epsilon^2 \sum_{i \in \mathbb{L}_\ell^c} \mathbb{I}(A_i \in \mathcal{I}). \tag{2}$$

We next provide an upper bound for the right-hand-side. Since $A$ has a bounded probability density function, the variance $\mathrm{Var}\{\mathbb{I}(A_i \in \mathcal{I})\}$ is upper bounded by $\sqrt{E\mathbb{I}(A_i \in \mathcal{I})} \le \bar{c}\sqrt{|\mathcal{I}|}$ for some universal constant $\bar{c} > 0$. It follows from Bernstein's inequality that

$$\mathrm{pr}\left\{\sum_{i \in \mathbb{L}_\ell^c} \mathbb{I}(A_i \in \mathcal{I}) - |\mathbb{L}_\ell^c|E\mathbb{I}(A \in \mathcal{I}) \ge t\right\} \le \exp\left(-\frac{t^2/2}{\bar{c}^2|\mathbb{L}_\ell^c||\mathcal{I}| + t/3}\right),$$

for any $t$ and $\mathcal{I}$. Set $t_{\mathcal{I}} = 6\max(\bar{c}\sqrt{n|\mathcal{I}|\log n}, |\mathcal{I}|\log n)$, the right-hand-side is upper bounded by $n^{-4}$. Since $m \asymp n$ and the number of intervals $\mathcal{I}$ in $\mathfrak{I}(m)$ is upper bounded by $m^2$, it follows from Bonferroni's inequality that

$$\mathrm{pr}\left[\bigcup_{\mathcal{I} \in \mathfrak{I}(m)} \left\{\sum_{i \in \mathbb{L}_\ell^c} \mathbb{I}(A_i \in \mathcal{I}) - |\mathbb{L}_\ell^c|E\mathbb{I}(A \in \mathcal{I}) \ge t_{\mathcal{I}}\right\}\right] \le m^2 n^{-4} = O(n^{-2}).$$

As such, with probability at least $1 - O(n^{-2})$, we have that $\sum_{i \in \mathbb{L}_\ell^c} \mathbb{I}(A_i \in \mathcal{I}) - |\mathbb{L}_\ell^c|E\mathbb{I}(A \in \mathcal{I}) \le t_{\mathcal{I}}$ uniformly for all $\mathcal{I}$, or equivalently, $\sum_{i \in \mathbb{L}_\ell^c} \mathbb{I}(A_i \in \mathcal{I}) \le |\mathbb{L}_\ell^c|\bar{c}|\mathcal{I}| + t_{\mathcal{I}}$. Consider a subset of intervals $\mathcal{I}$ with $|\mathcal{I}| \ge c\gamma_n$ for any constant $c > 0$. Under the given conditions on $\gamma_n$, we have

$$\sum_{i \in \mathbb{L}_\ell^c} \mathbb{I}(A_i \in \mathcal{I}) \le n\bar{c}^*|\mathcal{I}|, \quad \text{for any } \mathcal{I} \text{ such that } |\mathcal{I}| \ge c\gamma_n, \tag{3}$$

for some constant $\bar{c}^* > 0$. It follows from equation 2 that the following holds with probability at least $1 - O(n^{-2})$: for any $\mathcal{I} \in \mathfrak{I}(m)$ such that $|\mathcal{I}| \ge c\gamma_n$, we have

$$\sum_{i \in \mathbb{L}_\ell^c} \mathbb{I}(A_i \in \mathcal{I})|q_{\mathcal{I},0}(X_i) - q(X_i)|^2 \le \bar{c}^*\epsilon^2 n|\mathcal{I}|.$$

Set $\epsilon$ to $(n|\mathcal{I}|)^{-\beta/(2\beta+p)}$, it follows that

$$\sum_{i \in \mathbb{L}_\ell^c} \mathbb{I}(A_i \in \mathcal{I})|q_{\mathcal{I},0}(X_i) - q(X_i)|^2 \le \bar{c}^*(n|\mathcal{I}|)^{-2\beta/(2\beta+p)}(n|\mathcal{I}|). \tag{4}$$

$W_{\mathcal{I}}$ and $L_{\mathcal{I}}$ are upper bounded by $\bar{C}(n|\mathcal{I}|)^{p/(2\beta+p)}(\beta\log(n|\mathcal{I}|)/(2\beta + p) + 1)$ and $\bar{C}(\beta\log(n|\mathcal{I}|)/(2\beta + p) + 1)$, respectively. This completes the proof for Part 1.

*Part 2.* For the function class of deep neural networks $Q_{\mathcal{I}}$, we use $\theta_{\mathcal{I}}$ to denote the parameters in deep neural networks. This allows us to represent $\mathcal{Q}_{\mathcal{I}}$ as $\{q_{\mathcal{I}}(\bullet, \theta_{\mathcal{I}}) : \theta_{\mathcal{I}}\}$ We will apply the empirical process theory (see e.g., Van Der Vaart & Wellner 1996) to bound the stochastic error. Let $\widehat{\theta}_{\mathcal{I}}$ be the estimated parameter in $\widehat{q}_{\mathcal{I}}^{(\ell)}$. Define

$$\sigma^2(\mathcal{I}, \theta) = E\left\{\mathbb{I}(A \in \mathcal{I})|q_{\mathcal{I},0}(X) - q_{\mathcal{I},0}(X,\theta)|^2\right\},$$

for any $\theta$ and $\mathcal{I}$. Consider two separate cases, corresponding to $\sigma(\mathcal{I}, \widehat{\theta}_{\mathcal{I}}) \leq |\mathcal{I}|^{1/2}(n|\mathcal{I}|)^{-\beta/(2\beta+p)}$ and $\sigma(\mathcal{I}, \widehat{\theta}_{\mathcal{I}}) > |\mathcal{I}|^{1/2}(n|\mathcal{I}|)^{-\beta/(2\beta+p)}$, respectively. We focus our attentions on the latter class of intervals. In Part 3, we will show that for those intervals,

$$\sigma(\mathcal{I}, \widehat{\theta}_{\mathcal{I}}) \leq O(1)|\mathcal{I}|^{1/2}(n|\mathcal{I}|)^{-\beta/(2\beta+p)}\log^4 n,$$

for some universal constant $O(1)$. This implies that for any $\mathcal{I}$, we have

$$\sigma(\mathcal{I}, \widehat{\theta}_{\mathcal{I}}) \leq O(1)|\mathcal{I}|^{1/2}(n|\mathcal{I}|)^{-\beta/(2\beta+p)}\log^4 n. \tag{5}$$

We consider bounding a scaled version of the stochastic error,

$$\frac{1}{\sigma(\mathcal{I}, \widehat{\theta}_{\mathcal{I}})} \sum_{i \in \mathbb{L}_\ell^c} \mathbb{I}(A_i \in \mathcal{I})\{Y_i - q_{\mathcal{I},0}(X_i)\}\{\widehat{q}_{\mathcal{I}}^{(\ell)}(X_i) - q_{\mathcal{I},0}(X_i)\}.$$

Its absolute value can be upper bounded by

$$\mathbb{Z}(\mathcal{I}) \equiv \sup_{\theta} \left| \frac{1}{\sigma(\mathcal{I}, \theta)} \sum_{i \in \mathbb{L}_\ell^c} \mathbb{I}(A_i \in \mathcal{I})\{Y_i - q_{\mathcal{I},0}(X_i)\}\{q_{\mathcal{I},0}(X_i, \theta) - q_{\mathcal{I},0}(X_i)\} \right|,$$

where the supremum is taken over all $\theta$ such that $\sigma(\mathcal{I}, \theta) > |\mathcal{I}|^{1/2}(n|\mathcal{I}|)^{-\beta/(2\beta+p)}$.

For a given $\theta$, the empirical sum has zero mean. Under the boundedness assumption on $Y$, its variance is upper bounded by some universal constant. In addition, each quantity $\sigma^{-1}(\mathcal{I}, \theta)\mathbb{I}(A_i \in \mathcal{I})\{Y_i - q_{\mathcal{I},0}(X_i)\}\{q_{\mathcal{I},0}(X_i, \theta) - q_{\mathcal{I},0}(X_i)\}$ is upper bounded by $O(1)|\mathcal{I}|^{-1/2}(n|\mathcal{I}|)^{\beta/(2\beta+p)}$ for some universal constant $O(1)$. This allows us to apply the tail inequality developed by Massart et al. (2000) to bounded the empirical process. See also Theorem 2 of Adamczak et al. (2008). Specifically, for all $t > 0$ and $\mathcal{I}$ that satisfies $\sigma(\mathcal{I}, \widehat{\theta}_{\mathcal{I}}) > |\mathcal{I}|^{1/2}(n|\mathcal{I}|)^{-\beta/(2\beta+p)}$, we obtain with probability at least $1 - \exp(t)$ that

$$\mathbb{Z}(\mathcal{I}) \leq 2E\mathbb{Z}(\mathcal{I}) + \bar{c}\sqrt{tn} + t\bar{c}|\mathcal{I}|^{-1/2}(n|\mathcal{I}|)^{\beta/(2\beta+p)}, \tag{6}$$

for some constant $\bar{c} > 0$. By setting $t = 3\log n$, the probability $1 - \exp(t) = 1 - n^{-3}$. Notice that the number of intervals $\mathcal{I}$ is upper bounded by $O(n^2)$, under the condition that $m$ is proportional to $n$. By Bonferroni's inequality, we obtain that equation 6 holds with probability at least $1 - O(n^{-2})$ for any $\mathcal{I}$. Under the given condition on $\gamma_n$, for any interval $\mathcal{I}$ such that $|\mathcal{I}| \geq c\gamma_n$, the last term on the right-hand-side of equation 6 is $o(\sqrt{n})$. It follows that the following occurs with probability $1 - O(n^{-2})$,

$$\mathbb{Z}(\mathcal{I}) \leq 2E\mathbb{Z}(\mathcal{I}) + 2\bar{c}\sqrt{n\log n}, \tag{7}$$

for all $\mathcal{I}$ such that $|\mathcal{I}| \geq c\gamma_n$ and $\sigma(\mathcal{I}, \widehat{\theta}_{\mathcal{I}}) > |\mathcal{I}|^{1/2}(n|\mathcal{I}|)^{-\beta/(2\beta+p)}$.

We next provide an upper bound for $E\mathbb{Z}(\mathcal{I})$. Toward that end, we will apply the maximal inequality developed in Corollary 5.1 of Chernozhukov et al. (2014). We first observe that the class of empirical sum indexed by $\theta$ belongs to the VC subgraph class with VC-index upper bounded by $O(W_{\mathcal{I}}L_{\mathcal{I}}\log(W_{\mathcal{I}}))$. It follows that for any $\mathcal{I}$ such that $|\mathcal{I}| \geq c\gamma_n$, $\sigma(\mathcal{I}, \widehat{\theta}_{\mathcal{I}}) > |\mathcal{I}|^{1/2}(n|\mathcal{I}|)^{-\beta/(2\beta+p)}$,

$$E\mathbb{Z}(\mathcal{I}) \propto \sqrt{nW_{\mathcal{I}}L_{\mathcal{I}}\log(W_{\mathcal{I}})\log n} + W_{\mathcal{I}}L_{\mathcal{I}}\log(W_{\mathcal{I}})\log n.$$

Based on the upper bounds on $W_{\mathcal{I}}$ and $L_{\mathcal{I}}$ developed in Part 1, the right-hand-side is upper bounded by

$$O(1)(n|\mathcal{I}|)^{p/(4\beta+2p)}\sqrt{n\log^4 n} + O(1)|\mathcal{I}|^{-1/2}(n|\mathcal{I}|)^{p/(2\beta+p)}\log^4 n,$$

where $O(1)$ denotes some universal constant. It is of the order $O\{n^{1/2}(n|\mathcal{I}|)^{p/(4\beta+2p)}\log^4 n\}$. This yields that

$$E\mathbb{Z}(\mathcal{I}) \propto n^{1/2}(n|\mathcal{I}|)^{p/(4\beta+2p)}\log^4 n.$$

This together with equation 6 and equation 7 yields that with probability at least $1 - O(n^{-2})$, the scaled stochastic error is upper bounded by $n^{1/2}(n|\mathcal{I}|)^{p/(4\beta+2p)}\log^4 n$. As such, with probability at least $1 - O(n^{-2})$, we obtain that

$$\left|\sum_{i\in\mathbb{L}_\ell^c}\mathbb{I}(A_i\in\mathcal{I})\{Y_i - q_{\mathcal{I},0}(X_i)\}\{\widehat{q}_{\mathcal{I}}^{(\ell)}(X_i) - q_{\mathcal{I},0}(X_i)\}\right| \propto \sigma(\mathcal{I},\widehat{\theta}_{\mathcal{I}})n^{1/2}(n|\mathcal{I}|)^{p/(4\beta+2p)}\log^4 n,$$

for any $\mathcal{I}$ such that $|\mathcal{I}| \geq c\gamma_n$, $\sigma(\mathcal{I},\widehat{\theta}_{\mathcal{I}}) > |\mathcal{I}|^{1/2}(n|\mathcal{I}|)^{-\beta/(2\beta+p)}$. By Cauchy-Schwarz inequality, the left-hand-side can be further upper bounded by

$$\frac{n\sigma^2(\mathcal{I},\widehat{\theta}_{\mathcal{I}})}{4} + O(1)(n|\mathcal{I}|)^{p/(2\beta+p)}\log^8 n,$$

where $O(1)$ denotes some universal positive constant. This completes the proof for Part 2.

*Part 3.* Combining the results in Part 1 and Part 2, we obtain that for any $\mathcal{I}$ such that $|\mathcal{I}| \geq c\gamma_n$, $\sigma(\mathcal{I},\widehat{\theta}_{\mathcal{I}}) > |\mathcal{I}|^{1/2}(n|\mathcal{I}|)^{-\beta/(2\beta+p)}$,

$$\sum_{i\in\mathbb{L}_\ell^c}\mathbb{I}(A_i\in\mathcal{I})|q_{\mathcal{I},0}(X_i) - \widehat{q}_{\mathcal{I}}^{(\ell)}(X_i)|^2 \leq \frac{n\sigma^2(\mathcal{I},\widehat{\theta}_{\mathcal{I}})}{4} + O(1)(n|\mathcal{I}|)^{p/(2\beta+p)}\log^8 n,$$

with probability at least $1 - O(n^{-2})$. As for the left-hand-side, we notice that

$$\sum_{i\in\mathbb{L}_\ell^c}\mathbb{I}(A_i\in\mathcal{I})|q_{\mathcal{I},0}(X_i) - \widehat{q}_{\mathcal{I}}^{(\ell)}(X_i)|^2$$

$$\geq \quad |\mathbb{L}_\ell^c|\sigma^2(\mathcal{I},\widehat{\theta}_{\mathcal{I}}) - \left|\sum_{i\in\mathbb{L}_\ell^c}\mathbb{I}(A_i\in\mathcal{I})|q_{\mathcal{I},0}(X_i) - \widehat{q}_{\mathcal{I}}^{(\ell)}(X_i)|^2 - |\mathbb{L}_\ell^c|\sigma^2(\mathcal{I},\widehat{\theta}_{\mathcal{I}})\right|.$$

Using similar arguments in Part 2, we can show that the second line is upper bounded by $n\sigma^2(\mathcal{I},\widehat{\theta}_{\mathcal{I}})/8 + O(1)(n|\mathcal{I}|)^{p/(2\beta+p)}\log^8 n$, with probability at least $1 - O(n^{-2})$, for any $\mathcal{I}$ such that $|\mathcal{I}| \geq c\gamma_n$, $\sigma(\mathcal{I},\widehat{\theta}_{\mathcal{I}}) > |\mathcal{I}|^{1/2}(n|\mathcal{I}|)^{-\beta/(2\beta+p)}$. Since $\mathbb{L}_\ell^c \geq n/2$, we obtain

$$\left(\frac{1}{2} - \frac{1}{4} - \frac{1}{8}\right)\sigma^2(\mathcal{I},\widehat{\theta}_{\mathcal{I}}) = \frac{1}{8}\sigma^2(\mathcal{I},\widehat{\theta}_{\mathcal{I}}) \propto (n|\mathcal{I}|)^{-2\beta/(2\beta+p)}\log^8 n.$$

This yields the desired uniform upper bound for $\sigma^2(\mathcal{I},\widehat{\theta}_{\mathcal{I}})$. We thus obtain equation 5 holds with probability at least $1 - O(n^{-2})$.

Under the assumption that the density function $b(a|x)$ is uniformly bounded away from zero, we obtain

$$\sigma^2(\mathcal{I},\widehat{\theta}_{\mathcal{I}}) \leq c|\mathcal{I}|E|q_{\mathcal{I},0}(X) - \widehat{q}_{\mathcal{I}}^{(\ell)}(X)|^2,$$

for some constant $c > 0$. This assertion thus follows.

## E.2 Proof of Lemma E.2

The assertion can be proven in a similar manner as Part 2 of the proof of Lemma E.1. We omit the details to save space.

## E.3 Proof of Lemma E.3

Consider a given interval $\mathcal{I} \in \widehat{\mathcal{D}}^{(\ell)}$. Suppose $|\mathcal{I}| < c\gamma_n$. The value of the constant $c$ will be determined later. Then, for sufficiently large $n$, we can find some interval $\mathcal{I}' \in \mathfrak{I}(m) \cap \widehat{\mathcal{D}}^{(\ell)}$ that is

adjacent to $\mathcal{I}$. Thus, we have $\mathcal{I} \cup \mathcal{I}' \in \mathfrak{I}(m)$, and hence

$$\frac{1}{|\mathbb{L}_\ell^c|} \sum_{i \in \mathbb{L}_\ell^c} \mathbb{I}(A_i \in \mathcal{I})\{Y_i - \widehat{q}_{\mathcal{I}}^{(\ell)}(X_i)\}^2 + \frac{1}{|\mathbb{L}_\ell^c|} \sum_{i \in \mathbb{L}_\ell^c} \mathbb{I}(A_i \in \mathcal{I}')\{Y_i - \widehat{q}_{\mathcal{I}'}^{(\ell)}(X_i)\}^2 \quad (8)$$

$$\leq \quad \frac{1}{|\mathbb{L}_\ell^c|} \sum_{i \in \mathbb{L}_\ell^c} \mathbb{I}(A_i \in \mathcal{I} \cup \mathcal{I}')\{Y_i - \widehat{q}_{\mathcal{I} \cup \mathcal{I}'}^{(\ell)}(X_i)\}^2 - \gamma_n.$$

Notice that the left-hand-side of the above expression is nonnegative. It follows that

$$\gamma_n \leq \frac{1}{|\mathbb{L}_\ell^c|} \sum_{i \in \mathbb{L}_\ell^c} \mathbb{I}(A_i \in \mathcal{I} \cup \mathcal{I}')\{Y_i - \widehat{q}_{\mathcal{I} \cup \mathcal{I}'}^{(\ell)}(X_i)\}^2.$$

By definition, we have

$$\widehat{q}_{\mathcal{I} \cup \mathcal{I}'}^{(\ell)} = \arg\min_{q_\mathcal{I} \in \mathcal{Q}_\mathcal{I}} \frac{1}{n} \sum_{i \in \mathbb{L}_\ell^c} \mathbb{I}(A_i \in \mathcal{I} \cup \mathcal{I}')\{Y_i - q_\mathcal{I}(X_i)\}^2.$$

It follows that

$$\frac{1}{|\mathbb{L}_\ell^c|} \sum_{i \in \mathbb{L}_\ell^c} \mathbb{I}(A_i \in \mathcal{I} \cup \mathcal{I}')\{Y_i - \widehat{q}_{\mathcal{I} \cup \mathcal{I}'}^{(\ell)}(X_i)\}^2 \leq \frac{1}{|\mathbb{L}_\ell^c|} \sum_{i \in \mathbb{L}_\ell^c} \mathbb{I}(A_i \in \mathcal{I} \cup \mathcal{I}')\{Y_i - \widehat{q}_{\mathcal{I}'}^{(\ell)}(X_i)\}^2.$$

By equation 8, this further implies that

$$\frac{1}{|\mathbb{L}_\ell^c|} \sum_{i \in \mathbb{L}_\ell^c} \mathbb{I}(A_i \in \mathcal{I})\{Y_i - \widehat{q}_{\mathcal{I}}^{(\ell)}(X_i)\}^2 \leq \frac{1}{|\mathbb{L}_\ell^c|} \sum_{i \in \mathbb{L}_\ell^c} \mathbb{I}(A_i \in \mathcal{I})\{Y_i - \widehat{q}_{\mathcal{I}'}^{(\ell)}(X_i)\}^2 - \gamma_n,$$

and hence

$$\gamma_n \leq \frac{1}{|\mathbb{L}_\ell^c|} \sum_{i \in \mathbb{L}_\ell^c} \mathbb{I}(A_i \in \mathcal{I})\{Y_i - \widehat{q}_{\mathcal{I}'}^{(\ell)}(X_i)\}^2.$$

Under (A2), the function $\widehat{q}_{\mathcal{I}'}$ is uniformly upper bounded from above. It thus follows from Cauchy-Schwarz inequality that

$$\gamma_n \leq \frac{2}{|\mathbb{L}_\ell^c|} \sum_{i \in \mathbb{L}_\ell^c} \mathbb{I}(A_i \in \mathcal{I})\{Y_i^2 + \widehat{q}_{\mathcal{I}'}^2(X_i)\} \leq c_0 n^{-1} \sum_{i \in \mathbb{L}_\ell^c} \mathbb{I}(A_i \in \mathcal{I}),$$

for some constant $c_0 > 0$. Using similar arguments in showing equation 3, we can show that with probability at least $1 - O(n^{-2})$, the following evens hold for all $\mathcal{I} \in \mathfrak{I}(m)$,

$$n^{-1} \sum_{i \in \mathbb{L}_\ell^c} \mathbb{I}(A_i \in \mathcal{I}) \leq c_1(\sqrt{n^{-1}|\mathcal{I}| \log n} + |\mathcal{I}|),$$

for some constant $c_1 > 0$. The right-hand-side shall be larger than or equal to $\gamma_n$. Consequently, we have either $|\mathcal{I}| \geq c_2 \gamma_n$ or $|\mathcal{I}| \geq c_2 n \gamma_n^2 / \log n$ for some constant $c_2 > 0$. Under the given condition on $\gamma_n$, we obtain that $|\mathcal{I}| \geq c_2 \gamma_n$ for sufficiently large $n$. The proof is hence completed.

### E.4 Proof of Theorem 1

Since the number of folds $\mathcal{L}$ is a fixed integer. We will show the assertions in (i) and (ii) holds for each $\ell$, with probability at least $1 - O(n^{-2})$. The proof is divided into three parts. In Part 1, we show the consistency of the estimated change point locations and that $|\widehat{\mathcal{D}}^{(\ell)}| \geq |\mathcal{D}_0|$ with probability at least $1 - O(n^{-2})$. In Part 2, we prove that $|\widehat{\mathcal{D}}^{(\ell)}| = |\mathcal{D}_0|$ with probability at least $1 - O(n^{-2})$ and derive the rate of convergence of the estimated change point locations and the estimated function $Q$. In Part 3, we derive the rate of convergence for the value estimator.

*Part 1.* We first show the consistency of the estimated change-point locations. Assume $|\mathcal{D}_0| > 1$. Otherwise, the assertion $|\widehat{\mathcal{D}}^{(\ell)}| \geq |\mathcal{D}_0|$ trivially hold. Consider the partition $\mathcal{D} = \{[0, 1]\}$ which consists of a single interval and a zero function $Q$. By definition, we have

$$\sum_{\mathcal{I} \in \widehat{\mathcal{D}}^{(\ell)}} \left( \sum_{i \in \mathbb{L}_\ell^c} \mathbb{I}(A_i \in \mathcal{I})\{Y_i - \widehat{q}_{\mathcal{I}}^{(\ell)}(X_i)\}^2 \right) + |\mathbb{L}_\ell^c| \gamma_n |\widehat{\mathcal{D}}^{(\ell)}| \leq \sum_{i \in \mathbb{L}_\ell^c} Y_i^2 + |\mathbb{L}_\ell^c| \gamma_n.$$

Under the boundedness assumption on $Y$, we obtain that $|\mathbb{L}_\ell^c|\gamma_n|\widehat{\mathcal{D}}^{(\ell)}| \leq C_0(|\mathbb{L}_\ell^c| + \gamma_n)$ for some constant $C_0 > 0$ and hence

$$|\widehat{\mathcal{D}}^{(\ell)}| \leq 2C_0\gamma_n^{-1}, \tag{9}$$

for sufficiently large $n$, as $\gamma_n \to 0$.

Notice that

$$\sum_{\mathcal{I}\in\widehat{\mathcal{D}}^{(\ell)}} \sum_{i\in\mathbb{L}_\ell^c} \mathbb{I}(A_i \in \mathcal{I})\{Y_i - \widehat{q}_{\mathcal{I}}^{(\ell)}(X_i)\}^2 \geq \underbrace{\sum_{\mathcal{I}\in\widehat{\mathcal{D}}^{(\ell)}} \sum_{i\in\mathbb{L}_\ell^c} \mathbb{I}(A_i \in \mathcal{I})\{Y_i - q_{\mathcal{I},0}(X_i)\}^2}_{\eta_1^*}$$

$$+ \sum_{\mathcal{I}\in\widehat{\mathcal{D}}^{(\ell)}} \sum_{i\in\mathbb{L}_\ell^c} \mathbb{I}(A_i \in \mathcal{I})\{\widehat{q}_{\mathcal{I}}^{(\ell)}(X_i) - q_{\mathcal{I},0}(X_i)\}^2$$

$$-2 \sum_{\mathcal{I}\in\widehat{\mathcal{D}}^{(\ell)}} \left| \sum_{i\in\mathbb{L}_\ell^c} \mathbb{I}(A_i \in \mathcal{I})\{Y_i - q_{\mathcal{I},0}(X_i)\}\{\widehat{q}_{\mathcal{I}}^{(\ell)}(X_i) - q_{\mathcal{I},0}(X_i)\} \right|.$$

The second line is non-negative. Under Lemmas E.2 and E.3, the third line is lower bounded by $-C_1 \sum_{\mathcal{I}\in\widehat{\mathcal{D}}^{(\ell)}} (n|\mathcal{I}|)^{p/(p+2\beta)} \log^8 n$ for some constant $C_1 > 0$ with probability at least $1 - O(n^{-2})$. In view of equation 9, it can be further lower bounded by $-2C_0C_1\gamma_n^{-1}n^{p/(p+2\beta)} \log^8 n$. By equation 9 and the given condition on $\gamma_n$, the third line is $o(n)$. It follows that

$$\sum_{\mathcal{I}\in\widehat{\mathcal{D}}^{(\ell)}} \sum_{i\in\mathbb{L}_\ell^c} \mathbb{I}(A_i \in \mathcal{I})\{Y_i - \widehat{q}_{\mathcal{I}}^{(\ell)}(X_i)\}^2 \geq \eta_1^* + o(n), \tag{10}$$

with probability at least $1 - O(n^{-2})$.

Similar to equation 3, we can show that the following events occur with probability at least $1 - O(n^{-2})$,

$$\left| \frac{1}{|\mathbb{L}_\ell^c|} \sum_{i\in\mathbb{L}_\ell^c} \mathbb{I}(A_i \in \mathcal{I})\{Y_i - Q(X_i, A_i)\}\{Q(X_i, A_i) - q_{\mathcal{I},0}(X_i)\} \right| \tag{11}$$

$$\leq c_0 \left[ n^{-1/2}\sqrt{E\mathbb{I}(A \in \mathcal{I})\{Q(X, A) - q_{\mathcal{I},0}(X)\}^2 \log n} + n^{-1} \log n \right],$$

$$\left| \frac{1}{|\mathbb{L}_\ell^c|} \sum_{i\in\mathbb{L}_\ell^c} \mathbb{I}(A_i \in \mathcal{I})\{Q(X_i, A_i) - q_{\mathcal{I},0}(X_i)\}^2 - E\mathbb{I}(A \in \mathcal{I})|Q(X, A) - q_{\mathcal{I}}(X)|^2 \right| \tag{12}$$

$$\leq c_0 \left[ n^{-1/2}\sqrt{E\mathbb{I}(A \in \mathcal{I})\{Q(X, A) - q_{\mathcal{I},0}(X)\}^2 \log n} + n^{-1} \log n \right],$$

for some constant $c_0 > 0$. For any interval $\mathcal{I}$, the two upper bounds in equation 11 and equation 12 are $o(1)$.

It follows that

$$\eta_1^* = \sum_{\mathcal{I}\in\widehat{\mathcal{D}}^{(\ell)}} \sum_{i\in\mathbb{L}_\ell^c} \mathbb{I}(A_i \in \mathcal{I})\{Y_i - Q(X_i, A_i)\}^2 + \sum_{\mathcal{I}\in\widehat{\mathcal{D}}^{(\ell)}} \sum_{i\in\mathbb{L}_\ell^c} \mathbb{I}(A_i \in \mathcal{I})\{Q(X_i, A_i) - q_{\mathcal{I},0}(X_i)\}^2$$

$$+2 \sum_{\mathcal{I}\in\widehat{\mathcal{D}}^{(\ell)}} \sum_{i\in\mathbb{L}_\ell^c} \mathbb{I}(A_i \in \mathcal{I})\{Y_i - Q(X_i, A_i)\}\{Q(X_i, A_i) - q_{\mathcal{I},0}(X_i)\}$$

$$= \sum_{i\in\mathbb{L}_\ell^c} |Y_i - Q(X_i, A_i)|^2 + |\mathbb{L}_\ell^c| \sum_{\mathcal{I}\in\widehat{\mathcal{D}}^{(\ell)}} E\mathbb{I}(A \in \mathcal{I})|Q(X, A) - q_{\mathcal{I}}(X)|^2 + o(n),$$

with probability at least $1 - O(n^{-2})$. It follows from equation 10 that

$$\sum_{\mathcal{I}\in\widehat{\mathcal{D}}^{(\ell)}} \sum_{i\in\mathbb{L}_\ell^c} \mathbb{I}(A_i \in \mathcal{I})\{Y_i - \widehat{q}_{\mathcal{I}}^{(\ell)}(X_i)\}^2 \geq \underbrace{\sum_{i\in\mathbb{L}_\ell^c} |Y_i - Q(X_i, A_i)|^2}_{\eta_2^*}$$

$$+|\mathbb{L}_\ell^c| \sum_{\mathcal{I}\in\widehat{\mathcal{D}}^{(\ell)}} E\mathbb{I}(A \in \mathcal{I})|Q(X, A) - q_{\mathcal{I}}(X)|^2 + o(n), \tag{13}$$

with probability at least $1 - O(n^{-2})$.

Let us consider $\eta_2^*$. We observe that

$$\eta_2^* = \sum_{\mathcal{I} \in \mathcal{D}_0} \sum_{i \in \mathbb{L}_\ell^c} \mathbb{I}(A_i \in \mathcal{I}) |Y_i - q_{\mathcal{I},0}(X_i)|^2.$$

By the uniform approximation property of deep neural networks, there exists some $q_{\mathcal{I}}^* \in \mathcal{Q}_{\mathcal{I}}$ such that

$$\sum_{i \in \mathbb{L}_\ell^c} |q_{\mathcal{I},0}(X_i) - q_{\mathcal{I}}^*(X_i)|^2 \propto n(n|\mathcal{I}|)^{-2\beta/(2\beta+p)}.$$

See Part 1 of the proof of Lemma E.1 for details. Similar to equation 3, we can show that the following events occur with probability at least $1 - O(n^{-2})$,

$$\left| \frac{1}{|\mathbb{L}_\ell^c|} \sum_{i \in \mathbb{L}_\ell^c} \mathbb{I}(A_i \in \mathcal{I})\{Y_i - q_{\mathcal{I}}(X_i)\}\{q_{\mathcal{I}}(X_i) - q_{\mathcal{I}}^*(X_i)\} \right| \le \frac{c_0 \sqrt{|\mathcal{I}| \log n}}{\sqrt{n}} (n|\mathcal{I}|)^{-\beta/(2\beta+p)},$$

for some constant $c_0 > 0$ and any $\mathcal{I} \in \mathcal{D}_0$. It follows that

$$\eta_2^* - \sum_{\mathcal{I} \in \mathcal{D}_0} \sum_{i \in \mathbb{L}_\ell^c} \mathbb{I}(A_i \in \mathcal{I}) |Y_i - q_{\mathcal{I}}^*(X_i)|^2 \ge - \sum_{\mathcal{I} \in \mathcal{D}_0} \sum_{i \in \mathbb{L}_\ell^c} \mathbb{I}(A_i \in \mathcal{I}) |q_{\mathcal{I},0}(X_i) - q_{\mathcal{I}}^*(X_i)|^2$$

$$-2 \sum_{\mathcal{I} \in \mathcal{D}_0} \left| \sum_{i \in \mathbb{L}_\ell^c} \mathbb{I}(A_i \in \mathcal{I})\{Y_i - q_{\mathcal{I}}(X_i)\}\{q_{\mathcal{I}}(X_i) - q_{\mathcal{I}}^*(X_i)\} \right| \ge -\bar{c} n^{p/(2\beta+p)},$$

for some constant $\bar{c} > 0$. This together with equation 13 yields that

$$\sum_{\mathcal{I} \in \widehat{\mathcal{D}}^{(\ell)}} \sum_{i \in \mathbb{L}_\ell^c} \mathbb{I}(A_i \in \mathcal{I})\{Y_i - \widehat{q}_{\mathcal{I}}^{(\ell)}(X_i)\}^2 \ge \sum_{\mathcal{I} \in \mathcal{D}_0} \sum_{i \in \mathbb{L}_\ell^c} \mathbb{I}(A_i \in \mathcal{I}) |Y_i - q_{\mathcal{I}}^*(X_i)|^2$$

$$+ |\mathbb{L}_\ell^c| \sum_{\mathcal{I} \in \widehat{\mathcal{D}}^{(\ell)}} E\mathbb{I}(A \in \mathcal{I}) |Q(X, A) - q_{\mathcal{I},0}(X)|^2 + o(n) + O\{n^{p/(2\beta+p)}\}, \tag{14}$$

with probability at least $1 - O(n^{-2})$.

Let $K = |\mathcal{D}_0|$. For any integer $k$ such that $1 \le k \le K - 1$, let $\tau_{0,k}^*$ be the change point location that satisfies $\tau_{0,k}^* = i/m$ for some integer $i$ and that $|\tau_{0,k} - \tau_{0,k}^*| < m^{-1}$. Denoted by $\mathcal{D}^*$ the oracle partition formed by the change point locations $\{\tau_{0,k}^*\}_{k=1}^{K-1}$. Set $\tau_{0,0}^* = 0$, $\tau_{0,K}^* = 1$ and $q_{[\tau_{0,k-1}^*, \tau_{0,k}^*)}^{**} = q_{[\tau_{0,k-1}, \tau_{0,k})}^*$ for $1 \le k \le K - 1$. Let $\Delta_k = [\tau_{0,k-1}^*, \tau_{0,k}^*) \cap [\tau_{0,k-1}, \tau_{0,k})^c$ for $1 \le k \le K - 1$ and $\Delta_K = [\tau_{0,K-1}^*, 1] \cap [\tau_{0,K-1}, 1]^c$. The length of each interval $\Delta_k$ is at most $m^{-1}$. It follows that

$$\left( \sum_{\mathcal{I} \in \mathcal{D}^*} \left[ \sum_{i \in \mathbb{L}_\ell^c} \mathbb{I}(A_i \in \mathcal{I})\{Y_i - q_{\mathcal{I}}^{**}(X_i)\}^2 \right] + \gamma_n |\mathbb{L}_\ell^c| |\mathcal{D}^*| \right)$$

$$- \left( \sum_{\mathcal{I} \in \mathcal{D}_0} \left[ \sum_{i \in \mathbb{L}_\ell^c} \mathbb{I}(A_i \in \mathcal{I})\{Y_i - q_{\mathcal{I}}^*(X_i)\}^2 \right] + \gamma_n |\mathbb{L}_\ell^c| |\mathcal{D}_0| \right) \le 2 \sum_{k=1}^{K} \sum_{i \in \mathbb{L}_\ell^c} \mathbb{I}(A_i \in \Delta_k) \left\{ Y_i^2 + \sup_{\mathcal{I} \subseteq [0,1]} q_{\mathcal{I}}^{*2}(X_i) \right\}.$$

Since $Y$ is a bounded variable, $q_{\mathcal{I}}^*$ is uniformly bounded for any $\mathcal{I}$. The right-hand-side is upper bounded by $\sum_{k=1}^{K} \sum_{i \in \mathbb{L}_\ell^c} \mathbb{I}(A_i \in \Delta_k)$. Similar to equation 3, The later is upper bounded by $O(\log n)$, with probability at least $1 - O(n^{-2})$. It follows that

$$\left( \sum_{\mathcal{I} \in \mathcal{D}^*} \left[ \sum_{i \in \mathbb{L}_\ell^c} \mathbb{I}(A_i \in \mathcal{I})\{Y_i - q_{\mathcal{I}}^{**}(X_i)\}^2 \right] + \gamma_n |\mathbb{L}_\ell^c| |\mathcal{D}^*| \right) \tag{15}$$

$$- \left( \sum_{\mathcal{I} \in \mathcal{D}_0} \left[ \sum_{i \in \mathbb{L}_\ell^c} \mathbb{I}(A_i \in \mathcal{I})\{Y_i - q_{\mathcal{I}}^*(X_i)\}^2 \right] + \gamma_n |\mathbb{L}_\ell^c| |\mathcal{D}_0| \right) \le O(\log n),$$

with probability at least $1 - O(n^{-2})$. By definition,

$$\sum_{\mathcal{I} \in \widehat{\mathcal{D}}^{(\ell)}} \sum_{i \in \mathbb{L}_\ell^c} \mathbb{I}(A_i \in \mathcal{I})\{Y_i - \widehat{q}_{\mathcal{I}}^{(\ell)}(X_i)\}^2 + \gamma_n |\mathbb{L}_\ell^c| |\widehat{\mathcal{D}}^{(\ell)}|$$
$$\leq \sum_{\mathcal{I} \in \mathcal{D}^*} \sum_{i \in \mathbb{L}_\ell^c} \mathbb{I}(A_i \in \mathcal{I})\{Y_i - q_{\mathcal{I}}^{**}(X_i)\}^2 + \gamma_n |\mathbb{L}_\ell^c| |\mathcal{D}^*|. \tag{16}$$

Combining this together with equation 15 yields that

$$\sum_{\mathcal{I} \in \widehat{\mathcal{D}}^{(\ell)}} \sum_{i \in \mathbb{L}_\ell^c} \mathbb{I}(A_i \in \mathcal{I})\{Y_i - \widehat{q}_{\mathcal{I}}^{(\ell)}(X_i)\}^2 + \gamma_n |\mathbb{L}_\ell^c| |\widehat{\mathcal{D}}^{(\ell)}|$$
$$\leq \sum_{\mathcal{I} \in \mathcal{D}_0} \sum_{i \in \mathbb{L}_\ell^c} \mathbb{I}(A_i \in \mathcal{I})\{Y_i - q_{\mathcal{I}}^*(X_i)\}^2 + \gamma_n |\mathbb{L}_\ell^c| |\mathcal{D}_0| + O(\log n).$$

It follows from equation 14 and the condition $\gamma_n \to 0$ that

$$\sum_{\mathcal{I} \in \widehat{\mathcal{D}}^{(\ell)}} E \mathbb{I}(A \in \mathcal{I}) |Q(X, A) - q_{\mathcal{I},0}(X)|^2 = o(1), \tag{17}$$

with probability at least $1 - O(n^{-2})$. Under the event defined above, we show that $\max_{\tau \in J(\mathcal{D}_0)} \min_{\widehat{\tau} \in J(\widehat{\mathcal{D}}^{(\ell)})} |\widehat{\tau} - \tau| \leq \delta$ for any constant $\delta > 0$. This yields the consistency of our estimated change point locations.

Specifically, under the condition that $q_{\mathcal{I}_1,0} \neq q_{\mathcal{I}_2,0}$ for any adjacent $\mathcal{I}_1, \mathcal{I}_2 \in \mathcal{D}_0$, we have $E |q_{\mathcal{I}_1,0}(X) - q_{\mathcal{I}_2,0}(X)|^2 > 0$. Let $\delta_0$ denote the minimum distance between two change point locations. Since the change points are fixed, $\delta_0$ is a fixed positive value. For a given $0 < \delta < \delta_0$, suppose $\max_{\tau \in J(\mathcal{D}_0)} \min_{\widehat{\tau} \in J(\widehat{\mathcal{D}}^{(\ell)})} |\widehat{\tau} - \tau| > \delta$. Then there exists a change point $\tau_0$ and $\mathcal{I} \in \widehat{\mathcal{D}}^{(\ell)}$ such that $\tau_0 \in \mathcal{I}$, $|\mathcal{I}| \geq 2\delta$ and that $\min(|a - \tau_0|, |b - \tau_0|) \geq \delta$ where $a, b$ correspond to the endpoints of the interval $\mathcal{I}$. Under the event defined in equation 17, we have

$$E \mathbb{I}(A \in [a, b]) |Q(X, A) - q_{\mathcal{I},0}(X)|^2 = o(1). \tag{18}$$

Since $\delta_0 > \delta$, the conditional mean function $Q$ is a piecewise function of $A$ in the intervals $[a, \tau_0]$ and $[\tau_0, b]$. The left-hand-side thus equals

$$E \mathbb{I}(A \in [\tau_0, b)) |q_{[\tau_0, b],0}(X) - q_{\mathcal{I},0}(X)|^2 + E \mathbb{I}(A \in [a, \tau_0)) |q_{[a, \tau_0],0}(X) - q_{\mathcal{I},0}(X)|^2.$$

The function $q_{\mathcal{I},0}$ that minimizes the above objective is given by

$$\{E \mathbb{I}(A \in [a, b] | X)\}^{-1} [q_{[a, \tau_0],0}(X) E\{\mathbb{I}(A \in [a, \tau_0]) | X\} + q_{[\tau_0, b],0}(X) E\{\mathbb{I}(A \in [\tau_0, b)) | X\}].$$

Consequently, the left-hand-side of equation 18 is greater than or equal to

$$E\{\mathbb{I}(A \in [\tau_0, b)) | X\}\{\mathbb{I}(A \in [a, \tau_0]) | X\} |q_{[\tau_0, b],0}(X) - q_{[a, \tau_0],0}(X)|^2,$$

which is not to decay to zero since $\min(|a - \tau_0|, |b - \tau_0|) \geq \delta$ and that $q_{\mathcal{I}_1,0} \neq q_{\mathcal{I}_2,0}$ for any adjacent $\mathcal{I}_1, \mathcal{I}_2 \in \mathcal{D}_0$. This contradicts equation 18. As such, we obtain that $\max_{\tau \in J(\mathcal{D}_0)} \min_{\widehat{\tau} \in J(\widehat{\mathcal{D}}^{(\ell)})} |\widehat{\tau} - \tau| \leq \delta$ for any sufficiently small $\delta$. This yields the consistency of the estimated change point locations. It also implies that $|\widehat{\mathcal{D}}^{(\ell)}| \geq |\mathcal{D}_0|$ with probability at least $1 - O(n^{-2})$. This completes the proof of Part 1.

*Part 2.* In this part, we show $|\widehat{\mathcal{D}}^{(\ell)}| = |\mathcal{D}_0|$ with probability at least $1 - O(n^{-2})$ and derive the rate of convergence of the estimated change point locations. Similar to equation 14 and equation 15, with a more refined analysis (see Part 1 of the proof), we obtain that

$$\sum_{\mathcal{I} \in \widehat{\mathcal{D}}^{(\ell)}} \sum_{i \in \mathbb{L}_\ell^c} \mathbb{I}(A_i \in \mathcal{I})\{Y_i - \widehat{q}_{\mathcal{I}}(X_i)\}^2 \geq \sum_{\mathcal{I} \in \mathcal{D}^*} \sum_{i \in \mathbb{L}_\ell^c} \mathbb{I}(A_i \in \mathcal{I}) |Y_i - q_{\mathcal{I}}^{**}(X_i)|^2$$

$$+ |\mathbb{L}_\ell^c| \sum_{\mathcal{I} \in \widehat{\mathcal{D}}^{(\ell)}} E \mathbb{I}(A \in \mathcal{I}) |Q(X, A) - q_{\mathcal{I},0}(X)|^2 - C_1 |\widehat{\mathcal{D}}^{(\ell)}|^{\beta/(2p+\beta)} n^{p/(p+2\beta)} \log^8 n + O(n^{p/(2\beta+p)})$$

$$- 2c_0 |\mathbb{L}_\ell^c|^{1/2} \sum_{\mathcal{I} \in \widehat{\mathcal{D}}^{(\ell)}} \sqrt{E \mathbb{I}(A \in \mathcal{I})\{Q(X, A) - q_{\mathcal{I},0}(X)\}^2 \log n} - 2c_0 |\widehat{\mathcal{D}}^{(\ell)}| \log n.$$

with probability at least $1 - O(n^{-2})$. By Cauchy-Schwarz inequality, the third line is lower bounded by

$$-\frac{|\mathbb{L}_\ell^c|}{2} \sum_{\mathcal{I} \in \widehat{\mathcal{D}}^{(\ell)}} E\mathbb{I}(A \in \mathcal{I})|Q(X, A) - q_{\mathcal{I},0}(X)|^2 - 2(c_0 + c_0^2)|\widehat{\mathcal{D}}^{(\ell)}| \log n.$$

It follows that

$$\sum_{\mathcal{I} \in \widehat{\mathcal{D}}^{(\ell)}} \sum_{i \in \mathbb{L}_\ell^c} \mathbb{I}(A_i \in \mathcal{I})\{Y_i - \widehat{q}_{\mathcal{I}}(X_i)\}^2 \geq \sum_{\mathcal{I} \in \mathcal{D}^*} \sum_{i \in \mathbb{L}_\ell^c} \mathbb{I}(A_i \in \mathcal{I})|Y_i - q_{\mathcal{I}}^{**}(X_i)|^2$$

$$+ \frac{|\mathbb{L}_\ell^c|}{2} \sum_{\mathcal{I} \in \widehat{\mathcal{D}}^{(\ell)}} E\mathbb{I}(A \in \mathcal{I})|Q(X, A) - q_{\mathcal{I},0}(X)|^2 - C_1|\widehat{\mathcal{D}}^{(\ell)}|^{\beta/(2p+\beta)}n^{p/(p+2\beta)}\log^8 n$$

$$-2(c_0 + c_0^2)|\widehat{\mathcal{D}}^{(\ell)}|\log n + O(n^{p/(2\beta+p)}).$$

This together with equation 16 yields that

$$\frac{|\mathbb{L}_\ell^c|}{2} \sum_{\mathcal{I} \in \widehat{\mathcal{D}}^{(\ell)}} E\mathbb{I}(A \in \mathcal{I})|Q(X, A) - q_{\mathcal{I},0}(X)|^2 \leq C_1|\widehat{\mathcal{D}}^{(\ell)}|^{\beta/(2p+\beta)}n^{p/(p+2\beta)}\log^8 n$$

$$+O(n^{p/(2\beta+p)}) + n\gamma_n(|\mathcal{D}_0| - |\widehat{\mathcal{D}}^{(\ell)}|) + 2(c_0 + c_0^2)|\widehat{\mathcal{D}}^{(\ell)}|\log n.$$

Under the given condition on $\gamma_n$, we obtain that $|\widehat{\mathcal{D}}^{(\ell)}| \leq |\mathcal{D}_0|$. Combining this together with $|\widehat{\mathcal{D}}^{(\ell)}| \geq |\mathcal{D}_0|$, we obtain that $|\widehat{\mathcal{D}}^{(\ell)}| = |\mathcal{D}_0|$. As such, we have that

$$\sum_{\mathcal{I} \in \widehat{\mathcal{D}}^{(\ell)}} E\mathbb{I}(A \in \mathcal{I})|Q(X, A) - q_{\mathcal{I},0}(X)|^2 \propto n^{-2\beta/(p+2\beta)}\log^8 n.$$

Using similar arguments in establishing the consistency of the estimated change point locations, we can show that under the above event, we have that $\max_{\tau \in J(\mathcal{D}_0)} \min_{\widehat{\tau} \in J(\widehat{\mathcal{D}}^{(\ell)})} |\widehat{\tau} - \tau| \propto n^{-2\beta/(p+2\beta)}\log^8 n$. This completes the proof of this part.

*Part 3*. For any target policy $\pi$, we define a random policy $\pi_{\widehat{\mathcal{D}}^{(\ell)}}$ according to the partition $\widehat{\mathcal{D}}^{(\ell)}$ as follows:

$$\pi_{\widehat{\mathcal{D}}^{(\ell)}}(a|x) = \sum_{\mathcal{I} \subseteq \widehat{\mathcal{D}}^{(\ell)}} \mathbb{I}\{\pi(x) \in \mathcal{I}, a \in \mathcal{I}\}\frac{b(a|x)}{b(\mathcal{I}|x)},$$

where $b(\mathcal{I}|x)$ denotes the propensity score function $\text{pr}(A \in \mathcal{I}|X = x)$. Note that $\int_0^1 \pi_{\widehat{\mathcal{D}}^{(\ell)}}(a|x)da = \sum_{\mathcal{I} \subseteq \widehat{\mathcal{D}}^{(\ell)}} \mathbb{I}\{\pi(x) \in \mathcal{I}\} = 1$ for any $x$. Consequently, $\pi_{\widehat{\mathcal{D}}^{(\ell)}}$ is a valid random policy.

Since the behavior policy is known, the proposed doubly-robust estimator corresponds to an unbiased estimator for $\mathcal{L}^{-1}\sum_{\ell=1}^{\mathcal{L}} V(\pi_{\widehat{\mathcal{D}}^{(\ell)}})$. Using similar arguments in the causal inference literature on deriving the asymptotic property of doubly-robust estimators (Chernozhukov et al. 2017), we can show that

$$\widehat{V}(\pi) - \frac{1}{\mathcal{L}}\sum_{\ell=1}^{\mathcal{L}} V(\pi_{\widehat{\mathcal{D}}^{(\ell)}}) = O_p(n^{-1/2}).$$

It suffices to show $\mathcal{L}^{-1}\sum_{\ell=1}^{\mathcal{L}}\{V(\pi_{\widehat{\mathcal{D}}^{(\ell)}}) - V(\pi)\} = O_p\{n^{-2\beta/(2\beta+p)}\log^8 n\}$, or equivalently, $V(\pi_{\widehat{\mathcal{D}}^{(\ell)}}) - V(\pi) = O_p\{n^{-2\beta/(2\beta+p)}\log^8 n\}$.

Based on the results obtained in the first two parts, it follows from Cauchy-Schwarz inequality that

$$\sum_{\mathcal{I} \in \widehat{\mathcal{D}}^{(\ell)}} E\left[\mathbb{I}(A \in \mathcal{I})|Q(X, A) - \widehat{q}_{\mathcal{I}}^{(\ell)}(X)|^2|X\right] \leq 2\sum_{\mathcal{I} \in \widehat{\mathcal{D}}^{(\ell)}} E\mathbb{I}(A \in \mathcal{I})|Q(X, A) - q_{\mathcal{I},0}(X)|^2$$

$$+2\sum_{\mathcal{I} \in \widehat{\mathcal{D}}^{(\ell)}} E\left[\mathbb{I}(A \in \mathcal{I})|\widehat{q}_{\mathcal{I}}^{(\ell)}(X) - q_{\mathcal{I},0}(X)|^2|X\right] \propto n^{-2\beta/(p+2\beta)}\log^8 n. \tag{19}$$

Note that

$$V(\pi_{\widehat{\mathcal{D}}^{(\ell)}}) = E \int_{[0,1]} Q(X,a) \sum_{\mathcal{I} \subseteq \widehat{\mathcal{D}}^{(\ell)}} \mathbb{I}\{\pi(X) \in \mathcal{I}, a \in \mathcal{I}\} \frac{b(a|X)}{b(\mathcal{I}|X)} da$$

$$= \sum_{\mathcal{I}_0 \in \mathcal{D}_0} E q_{\mathcal{I}_0}(X) \sum_{\mathcal{I} \subseteq \widehat{\mathcal{D}}^{(\ell)}} \mathbb{I}\{\pi(X) \in \mathcal{I}\} \frac{b(\mathcal{I} \cap \mathcal{I}_0|X)}{b(\mathcal{I}|X)}.$$

Similarly, we can show

$$V(\pi) = \sum_{\mathcal{I}_0 \in \mathcal{D}_0} E q_{\mathcal{I}_0}(X) \mathbb{I}\{\pi(X) \in \mathcal{I}_0\}.$$

It follows that

$$|V(\pi_{\widehat{\mathcal{D}}^{(\ell)}}) - V(\pi)| \leq \sum_{\mathcal{I}_0 \in \mathcal{D}_0} E |q_{\mathcal{I}_0}(X)| \left| \mathbb{I}\{\pi(X) \in \mathcal{I}_0\} - \sum_{\mathcal{I} \subseteq \widehat{\mathcal{D}}^{(\ell)}} \mathbb{I}\{\pi(X) \in \mathcal{I}\} \frac{b(\mathcal{I} \cap \mathcal{I}_0|X)}{b(\mathcal{I}|X)} \right|.$$

As $q_{\mathcal{I}_0}$ is uniformly bounded, the left-hand-side is upper bounded by

$$\sum_{\mathcal{I}_0 \in \mathcal{D}_0} E \left| \mathbb{I}\{\pi(X) \in \mathcal{I}_0\} - \sum_{\mathcal{I} \subseteq \widehat{\mathcal{D}}^{(\ell)}} \mathbb{I}\{\pi(X) \in \mathcal{I}\} \frac{b(\mathcal{I} \cap \mathcal{I}_0|X)}{b(\mathcal{I}|X)} \right|. \tag{20}$$

Based on the results obtained in Part 2, for each $\mathcal{I}_0 \in \mathcal{D}_0$, there exists some $\mathcal{I}_0^{(\ell)}$ where the Lebesgue measure of the difference $\mathcal{I}_0 \cap (\mathcal{I}_0^{(\ell)})^c + \mathcal{I}_0^c \cap \mathcal{I}_0^{(\ell)}$ is upper bounded by $O\{n^{-2\beta/(2\beta+p)} \log^8 n\}$, with probability at least $1 - O(n^{-2})$. The upper bound in equation 20 is $O\{n^{-2\beta/(2\beta+p)} \log^8 n\}$, under the positivity assumption and the assumption that $\text{pr}(\pi(X) \in [\tau_0 - \epsilon, \tau_0 + \epsilon]) = O(\epsilon)$ for any $\tau_0 \in J(\mathcal{D}_0)$ and sufficiently small $\epsilon > 0$. This completes the proof.

### E.5   Proof of Theorem 2

We break the proof into two parts. In Part 1, we introduce an auxiliary lemma and present its proof. In Part 2, we derive the convergence rate of the proposed value estimator.

*Part 1.* We first introduce the following lemma.

**Lemma E.4** *For any interval $\mathcal{I} \in \mathfrak{I}(m)$ with $|\mathcal{I}| \gg \gamma_n$ and any interval $\mathcal{I}' \in \widehat{\mathcal{D}}^{(\ell)}$ with $\mathcal{I} \subseteq \mathcal{I}'$, we have with probability approaching $1$ that*

$$E|q_{\mathcal{I},0}(X) - q_{\mathcal{I}',0}(X)|^2 \leq \bar{C} |\mathcal{I}|^{-1} \gamma_n,$$

*for some constant $\bar{C} > 0$.*

We next prove Lemma E.4. For a given interval $\mathcal{I}' \in \widehat{\mathcal{D}}^{(\ell)}$, the set of intervals $\mathcal{I}$ considered in Lemma E.4 can be classified into the following three categories.

*Category 1:* $\mathcal{I} = \mathcal{I}'$. It is immediate to see that $q_{\mathcal{I}} = q_{\mathcal{I}'}$ and the assertion automatically holds.

*Category 2:* There exists another interval $\mathcal{I}^* \in \mathfrak{I}(m)$ that satisfies $\mathcal{I}' = \mathcal{I}^* \cup \mathcal{I}$. Notice that the partition $\widehat{\mathcal{D}}^{(\ell)*} = \widehat{\mathcal{D}}^{(\ell)} \cup \{\mathcal{I}^*\} \cup \mathcal{I} - \{\mathcal{I}'\}$ forms another partition. By definition, we have

$$\frac{1}{|\mathbb{L}_\ell^c|} \sum_{i \in \mathbb{L}_\ell^c} \sum_{\mathcal{I}_0 \in \widehat{\mathcal{D}}^{(\ell)*}} \mathbb{I}(A_i \in \mathcal{I}_0)\{Y_i - \widehat{q}_{\mathcal{I}_0}(X_i)\}^2 + \gamma_n |\widehat{\mathcal{D}}^{(\ell)*}|$$

$$\geq \quad \frac{1}{|\mathbb{L}_\ell^c|} \sum_{i \in \mathbb{L}_\ell^c} \sum_{\mathcal{I}_0 \in \widehat{\mathcal{D}}^{(\ell)}} \mathbb{I}(A_i \in \mathcal{I}_0)\{Y_i - \widehat{q}_{\mathcal{I}_0}(X_i)\}^2 + \gamma_n |\widehat{\mathcal{D}}^{(\ell)}|,$$

and hence

$$\frac{1}{|\mathbb{L}_\ell^c|} \sum_{i \in \mathbb{L}_\ell^c} \mathbb{I}(A_i \in \mathcal{I})\{Y_i - \widehat{q}_{\mathcal{I}}(X_i)\}^2 + \frac{1}{|\mathbb{L}_\ell^c|} \sum_{i \in \mathbb{L}_\ell^c} \mathbb{I}(A_i \in \mathcal{I}^*)\{Y_i - \widehat{q}_{\mathcal{I}^*}(X_i)\}^2$$

$$\geq \frac{1}{|\mathbb{L}_\ell^c|} \sum_{i \in \mathbb{L}_\ell^c} \mathbb{I}(A_i \in \mathcal{I}')\{Y_i - \widehat{q}_{\mathcal{I}'}(X_i)\}^2 - \gamma_n.$$

It follows from the definition of $\widehat{q}_{\mathcal{I}^*}$ that

$$\frac{1}{|\mathbb{L}_\ell^c|}\sum_{i\in\mathbb{L}_\ell^c}\mathbb{I}(A_i\in\mathcal{I}^*)\{Y_i-\widehat{q}_{\mathcal{I}^*}(X_i)\}^2\leq\frac{1}{|\mathbb{L}_\ell^c|}\sum_{i\in\mathbb{L}_\ell^c}\mathbb{I}(A_i\in\mathcal{I}^*)\{Y_i-\widehat{q}_{\mathcal{I}'}(X_i)\}^2.$$

Therefore, we obtain

$$\frac{1}{|\mathbb{L}_\ell^c|}\sum_{i\in\mathbb{L}_\ell^c}\mathbb{I}(A_i\in\mathcal{I})\{Y_i-\widehat{q}_{\mathcal{I}}(X_i)\}^2\geq\frac{1}{|\mathbb{L}_\ell^c|}\sum_{i\in\mathbb{L}_\ell^c}\mathbb{I}(A_i\in\mathcal{I})\{Y_i-\widehat{q}_{\mathcal{I}'}(X_i)\}^2-\gamma_n. \tag{21}$$

*Category 3:* There exist two intervals $\mathcal{I}^*,\mathcal{I}^{**}\in\mathfrak{I}(m)$ that satisfy $\mathcal{I}'=\mathcal{I}^*\cup\mathcal{I}\cup\mathcal{I}^{**}$. Using similar arguments in proving equation 21, we can show that

$$\frac{1}{|\mathbb{L}_\ell^c|}\sum_{i\in\mathbb{L}_\ell^c}\mathbb{I}(A_i\in\mathcal{I})\{Y_i-\widehat{q}_{\mathcal{I}}(X_i)\}^2\geq\frac{1}{|\mathbb{L}_\ell^c|}\sum_{i\in\mathbb{L}_\ell^c}\mathbb{I}(A_i\in\mathcal{I})\{Y_i-\widehat{q}_{\mathcal{I}'}(X_i)\}^2-2\gamma_n.$$

Hence, regardless of whether $\mathcal{I}$ belongs to Category 2, or it belongs to Category 3, we have

$$\frac{1}{|\mathbb{L}_\ell^c|}\sum_{i\in\mathbb{L}_\ell^c}\mathbb{I}(A_i\in\mathcal{I})\{Y_i-\widehat{q}_{\mathcal{I}}(X_i)\}^2\geq\frac{1}{|\mathbb{L}_\ell^c|}\sum_{i\in\mathbb{L}_\ell^c}\mathbb{I}(A_i\in\mathcal{I})\{Y_i-\widehat{q}_{\mathcal{I}'}(X_i)\}^2-2\gamma_n. \tag{22}$$

Notice that for any interval $\mathcal{I}_0$,

$$\frac{1}{|\mathbb{L}_\ell^c|}\sum_{i\in\mathbb{L}_\ell^c}\mathbb{I}(A_i\in\mathcal{I}_0)\{Y_i-\widehat{q}_{\mathcal{I}_0}(X_i)\}^2-E[\mathbb{I}(A\in\mathcal{I}_0)\{Y-\widehat{q}_{\mathcal{I}_0}(X)\}^2|\{O_i\}_{i\in\mathbb{L}_\ell^c}]$$

$$=\frac{1}{|\mathbb{L}_\ell^c|}\sum_{i\in\mathbb{L}_\ell^c}\mathbb{I}(A_i\in\mathcal{I}_0)\{\widehat{q}_{\mathcal{I}_0}(X_i)-q_{\mathcal{I}_0,0}(X_i)\}\{q_{\mathcal{I},0}(X_i)-\widehat{q}_{\mathcal{I}_0,0}(X_i)\}^2$$

$$+\frac{1}{|\mathbb{L}_\ell^c|}\sum_{i\in\mathbb{L}_\ell^c}\mathbb{I}(A_i\in\mathcal{I}_0)\{Y_i-\widehat{q}_{\mathcal{I}_0}(X_i)\}^2-E[\mathbb{I}(A\in\mathcal{I}_0)\{\widehat{q}_{\mathcal{I}_0}(X_i)-\widehat{q}_{\mathcal{I}_0}(X)\}^2|\{O_i\}_{i\in\mathbb{L}_\ell^c}].$$

Using similar arguments in bounding the stochastic error term in Part 2 of the proof of Lemma E.1, we can show with probability approaching 1 that the right-hand-side is of the order $O\{n^{-2\beta/(2\beta+p)}\log^8 n\}$, for any $\mathcal{I}_0\in\mathfrak{I}(m)$. As such, we obtain with probability approaching 1 that

$$\frac{1}{|\mathbb{L}_\ell^c|}\sum_{i\in\mathbb{L}_\ell^c}\mathbb{I}(A_i\in\mathcal{I})\{Y_i-\widehat{q}_{\mathcal{I}}(X_i)\}^2=E[\mathbb{I}(A\in\mathcal{I})\{Y-\widehat{q}_{\mathcal{I}}(X)\}^2|\{O_i\}_{i\in\mathbb{L}_\ell^c}]$$

$$+O(1)|\mathcal{I}|(n|\mathcal{I}|)^{-2\beta/(2\beta+p)}\log^8 n,$$

$$\frac{1}{|\mathbb{L}_\ell^c|}\sum_{i\in\mathbb{L}_\ell^c}\mathbb{I}(A_i\in\mathcal{I})\{Y_i-\widehat{q}_{\mathcal{I}'}(X_i)\}^2=E[\mathbb{I}(A\in\mathcal{I})\{Y-\widehat{q}_{\mathcal{I}'}(X)\}^2|\{O_i\}_{i\in\mathbb{L}_\ell^c}]$$

$$+O(1)|\mathcal{I}|(n|\mathcal{I}|)^{-2\beta/(2\beta+p)}\log^8 n,$$

where $O(1)$ denotes some universal positive constant. Combining these together with equation 22 yields

$$E[\mathbb{I}(A\in\mathcal{I})\{Y-\widehat{q}_{\mathcal{I}}(X)\}^2|\{O_i\}_{i\in\mathbb{L}_\ell^c}]\geq E[\mathbb{I}(A\in\mathcal{I})\{Y-\widehat{q}_{\mathcal{I}'}(X)\}^2|\{O_i\}_{i\in\mathbb{L}_\ell^c}]$$

$$-2\gamma_n+O(1)|\mathcal{I}|(n|\mathcal{I}|)^{-2\beta/(2\beta+p)}\log^8 n,$$

for any $\mathcal{I}$ and $\mathcal{I}'$, with probability approaching 1. Note that $q_{\mathcal{I},0}$ satisfies $E[\mathbb{I}(A\in\mathcal{I})\{Y-q_{\mathcal{I},0}(X)\}|X]=0$. We have

$$E[\mathbb{I}(A\in\mathcal{I})\{q_{\mathcal{I},0}(X)-\widehat{q}_{\mathcal{I}}(X)\}^2|\{O_i\}_{i\in\mathbb{L}_\ell^c}]\geq E[\mathbb{I}(A\in\mathcal{I})\{q_{\mathcal{I},0}(X)-\widehat{q}_{\mathcal{I}'}(X)\}^2|\{O_i\}_{i\in\mathbb{L}_\ell^c}]$$

$$-2\gamma_n+O(1)|\mathcal{I}|(n|\mathcal{I}|)^{-2\beta/(2\beta+p)}\log^8 n.$$

Consider the first term on the right-hand-side. Note that

$$E[\mathbb{I}(A\in\mathcal{I})\{q_{\mathcal{I},0}(X)-\widehat{q}_{\mathcal{I}'}(X)\}^2|\{O_i\}_{i\in\mathbb{L}_\ell^c}]=E[\mathbb{I}(A\in\mathcal{I})\{q_{\mathcal{I},0}(X)-q_{\mathcal{I}'}(X)\}^2|\{O_i\}_{i\in\mathbb{L}_\ell^c}]$$

$$+E[\mathbb{I}(A\in\mathcal{I})\{\widehat{q}_{\mathcal{I}'}(X)-q_{\mathcal{I}',0}(X)\}^2|\{O_i\}_{i\in\mathbb{L}_\ell^c}]$$

$$-2E[\mathbb{I}(A\in\mathcal{I})\{q_{\mathcal{I},0}(X)-q_{\mathcal{I}',0}(X)\}\{\widehat{q}_{\mathcal{I}'}(X)-q_{\mathcal{I}',0}(X)\}|\{O_i\}_{i\in\mathbb{L}_\ell^c}].$$

By Cauchy-Schwarz inequality, the last term on the right-hand-side can be lower bounded by

$$-\frac{1}{2}E[\mathbb{I}(A \in \mathcal{I})\{q_{\mathcal{I},0}(X) - q_{\mathcal{I}',0}(X)\}^2|\{O_i\}_{i \in \mathbb{L}_\ell^c}] - 2E[\mathbb{I}(A \in \mathcal{I})\{\hat{q}_{\mathcal{I}'}(X) - q_{\mathcal{I}',0}(X)\}^2|\{O_i\}_{i \in \mathbb{L}_\ell^c}].$$

It follows that

$$E[\mathbb{I}(A \in \mathcal{I})\{q_{\mathcal{I},0}(X) - \hat{q}_{\mathcal{I}'}(X)\}^2|\{O_i\}_{i \in \mathbb{L}_\ell^c}] \geq \frac{1}{2}E[\mathbb{I}(A \in \mathcal{I})\{q_{\mathcal{I},0}(X) - q_{\mathcal{I}',0}(X)\}^2|\{O_i\}_{i \in \mathbb{L}_\ell^c}]$$
$$- 3E[\mathbb{I}(A \in \mathcal{I})\{\hat{q}_{\mathcal{I}'}(X) - q_{\mathcal{I}',0}(X)\}^2|\{O_i\}_{i \in \mathbb{L}_\ell^c}],$$

and hence

$$\frac{1}{2}E[\mathbb{I}(A \in \mathcal{I})\{q_{\mathcal{I},0}(X) - q_{\mathcal{I}',0}(X)\}^2|\{O_i\}_{i \in \mathbb{L}_\ell^c}] - 2\gamma_n + O(1)|\mathcal{I}|(n|\mathcal{I}|)^{-2\beta/(2\beta+p)}\log^8 n$$
$$\leq E[\mathbb{I}(A \in \mathcal{I})\{q_{\mathcal{I},0}(X) - \hat{q}_{\mathcal{I}'}(X)\}^2|\{O_i\}_{i \in \mathbb{L}_\ell^c}] + 3E[\mathbb{I}(A \in \mathcal{I})\{q_{\mathcal{I}',0}(X) - \hat{q}_{\mathcal{I}'}(X)\}^2|\{O_i\}_{i \in \mathbb{L}_\ell^c}].$$

By Lemma E.1, Lemma E.3 and the positivity assumption, the right-hand-side is upper bounded by $O(1)|\mathcal{I}|(n|\mathcal{I}|)^{-2\beta/(p+2\beta)}\log^8 n$ for some universal positive constant $O(1)$, with probability approaching 1. We obtain with probability approaching 1 that

$$E[\mathbb{I}(A \in \mathcal{I})\{q_{\mathcal{I}}(X) - q_{\mathcal{I}'}(X)\}^2|\{O_i\}_{i \in \mathbb{L}_\ell^c}] = 4\gamma_n + O(1)|\mathcal{I}|(n|\mathcal{I}|)^{-2\beta/(2\beta+p)}\log^8 n,$$

uniformly for any $\mathcal{I}$ and $\mathcal{I}'$, or equivalently,

$$E\left[\frac{b(\mathcal{I}|X)}{|\mathcal{I}|}\{q_{\mathcal{I}}(X) - q_{\mathcal{I}'}(X)\}^2|\{O_i\}_{i \in \mathbb{L}_\ell^c}\right] = \frac{4\gamma_n}{|\mathcal{I}|} + O(1)(n|\mathcal{I}|)^{-2\beta/(2\beta+p)}\log^8 n.$$

By the positivity assumption, we have with probability approaching 1 that

$$E[\{q_{\mathcal{I}}(X) - q_{\mathcal{I}'}(X)\}^2|\{O_i\}_{i \in \mathbb{L}_\ell^c}] = O(\gamma_n|\mathcal{I}|^{-1}) + O\{(n|\mathcal{I}|)^{-2\beta/(2\beta+p)}\log^8 n\},$$

uniformly for any $\mathcal{I}$ and $\mathcal{I}'$. The proof is hence completed by noting that $\gamma_n$ is at least of the order $O(n^{-2\beta}/(2\beta+p))\log^8 n$.

*Part 2.* Consider the bias of the proposed estimator first. Similar to Part 3 of the proof of Theorem 1, the bias is given by $\mathcal{L}^{-1}\sum_{\ell=1}^{\mathcal{L}} V(\pi_{\hat{\mathcal{D}}^{(\ell)}}) - V(\pi)$. By definition,

$$V(\pi_{\hat{\mathcal{D}}^{(\ell)}}) - V(\pi) = \sum_{\mathcal{I} \in \hat{\mathcal{D}}^{(\ell)}} \int_{\mathcal{I}} EQ(X,a)\mathbb{I}(\pi(X) \in \mathcal{I})\frac{b(a|X)}{b(\mathcal{I}|X)}da - EQ\{X, \pi(X)\}$$

$$= \sum_{\mathcal{I} \in \hat{\mathcal{D}}^{(\ell)}} \int_{\mathcal{I}} E\{Q(X,a) - Q\{X, \pi(X)\}\}\mathbb{I}(\pi(X) \in \mathcal{I})\frac{b(a|X)}{b(\mathcal{I}|X)}da$$

$$= \sum_{\mathcal{I}' \in \hat{\mathcal{D}}^{(\ell)}} E\{q_{\mathcal{I},0}(X) - Q\{X, \pi(X)\}\}\mathbb{I}(\pi(X) \in \mathcal{I}).$$

It follows that

$$|V(\pi_{\hat{\mathcal{D}}^{(\ell)}}) - V(\pi)| \leq \sup_{\mathcal{I}' \in \hat{\mathcal{D}}^{(\ell)}, a \in \mathcal{I}'} E|Q(X,a) - q_{\mathcal{I}'}(X)|. \tag{23}$$

For any $\mathcal{I}' \in \hat{\mathcal{D}}^{(\ell)}$. Consider two separate cases, corresponding to $|\mathcal{I}'| \leq \gamma_n^{1/3}$ and $|\mathcal{I}'| > \gamma_n^{1/3}$, respectively.

In Case 1, it follows from the Lipschitz property of the conditional mean function $Q$ that $|Q(x, a_1) - Q(x, a_2)| \leq L\gamma_n^{1/3}$, for any $a_1, a_2 \in \mathcal{I}'$ and $x$. By definition, the function $q_{\mathcal{I}'}$ can be represented as $q_{\mathcal{I}'}(x) = \int_{\mathcal{I}'} Q(x,a)\omega(a,x)da$ for some weight function $\omega$ such that $\int_{\mathcal{I}'} \omega(a,x)da = 1$. It follows that the right-hand-side of equation 23 is upper bounded by $L\gamma_n^{1/3}$.

In Case 2, for any $a \in \mathcal{I}'$, we can find an interval $\mathcal{I} \subseteq \mathcal{I}'$, $a \in \mathcal{I}$ with length proportional to $\gamma_n^{1/3}$. Using similar arguments in Case 1, we can show that $|Q(x,a) - q_{\mathcal{I},0}(x)| \leq L\gamma_n^{1/3}$. By Lemma E.4 and the Cauchy-Schwarz inequality, we have

$$E|q_{\mathcal{I},0}(X) - q_{\mathcal{I}',0}(X)| \leq \sqrt{\bar{C}\gamma_n^{2/3}} = \bar{C}^{1/2}\gamma_n^{1/3},$$

with probability approaching 1. It follows that the right-hand-side of equation 23 is upper bounded by $(L + \sqrt{\overline{C}})\gamma_n^{1/3}$, with probability approaching 1.

As such, the bias of the proposed estimator is upper bounded by

$$(L + \sqrt{\overline{C}})\gamma_n^{1/3},$$

with probability approaching 1.

We next consider the standard deviation of our estimator. The proposed estimator is can be represented by $\mathcal{L}^{-1} \sum_{\ell=1}^{\mathcal{L}} \widehat{V}^\ell(\pi)$ where $\widehat{V}^\ell(\pi)$ is the value estimator constructed based on the samples in $\{O_i\}_{i \in \mathbb{L}_\ell}$. Since the propensity score function is known to us, each $\widehat{V}^\ell(\pi)$ is unbiased to $V(\pi_{\widehat{\mathcal{D}}^{(\ell)}})$. Under the positivity assumption and the boundedness assumption on $Y$ and $\widehat{q}_{\mathcal{I}}$, the variance of $\widehat{V}^\ell(\pi)$ is upper bounded by $|\mathbb{L}_\ell|^{-1} \inf_{\mathcal{I} \in \widehat{\mathcal{D}}^{(\ell)}} |\mathcal{I}|^{-1}$. By Lemma E.3, it is upper bounded by $O(n^{-1}\gamma_n^{-1})$. As such, the standard deviation of our estimator is upper bounded by $O(n^{-1}\gamma_n^{-1})$.

As such, the convergence rate is given by $O_p(\gamma_n^{1/3} + n^{-1/2}\gamma_n^{-1/2})$. By setting $\gamma_n = n^{-3/5}$, the rate is given by $O_p(n^{-1/5})$. The proof is hence completed.