# OpenReview forum: "Deep Jump Learning for Off-Policy Evaluation in Continuous Treatment Settings"
_NeurIPS.cc/2021/Conference — NeurIPS 2021 Poster_

### Official Review · Reviewer_Ru8S · 2021-07-15

**Rating:** 7
**Confidence:** 3

**Summary:**

This paper introduces an alternative to doubly robust policy evaluation using kernel methods, which are limited by an assumption that the estimated state-action value (Q) function is smooth and has a second derivative. The proposed estimator, built from multi-scale change-point detection, is able to account for possible discontinuities in the Q function in a way that both reduces bias and variance. The proposed estimation approach is effectively shown to improve over prior methods on a suite of synthetic experiments as well as a real-world dataset.

**Limitations And Societal Impact:**

The limitation that I foresee with the proposed DJL estimation approach is that it requires extensive computational approximations of the piecewise Q function, the partitioning of the treatment space, as well as the propensity estimator. The authors conveniently ignore the complexity of learning these separate MLP functions and do not address the time or computational complexity required to implement their approach. While likely scaling better than the competing kernel methods, I feel that the authors should devote some space in their response to address the challenges and complexity of training the various estimators used to form their proposed DJL approach. This should also be included in the paper to more completely present the method.

Another limitation, (possibly should be mentioned in the Quality/Clarity section above) is that there is no clear discussion surrounding how one would be able to determine what modeling assumption to use when selecting between a piecewise of continuous Q function approximation. The relationship and apparent changes needed to the proposed DJL in the face of one over the other wasn't very clear.

**Main Review:**

#### **High-level evaluation**
Altogether, I appreciated the work presented in this paper. Off-policy evaluation is a difficult challenge facing real-world implementation of decision support algorithms. While I generally view the work positively and consider it highly, there are several areas where the paper could better describe the details of the technical foundations underlying the proposed DJL estimation approach. For this reason, I am a little hesitant to recommend that this paper be published. I have posed several questions below about some points that were unclear to me as a reviewer and someone who would potentially use this approach. I am quite ready to raise my score if the authors are able to satisfactorily address the concerns I raise in the Quality/Clarity section below as well as in the Limitations field below.

#### **Originality**
The paper points out an important limitation of recent continuous-action Doubly Robust OPE approaches in that they rely on the existence of the second derivative of the action-value function Q. The paper rescopes the reliance on the continuity of Q by estimating the expected outcome as a piecewise constant function, using a parametrized form of change-point detection. This function is adaptively discretized so that it accounts for local discontinuities in the estimated value function.

#### **Quality and Clarity**
In the introduction, the limitations of kernel-based methods are referred to but not specified. Ideally, the paper would more directly list these specific limitations as well as how the proposed DJL addresses them. The paper would be easier to follow if these were made more clear.

Some questions that remain after reading Section 2.1:
Are the feature-treatment-outcome observations fixed in time? In other words, is the problem set up to only consider single step treatment scenarios? As a follow-up, what are the limitations of DJL when considering the extension to sequential decision making? Does the formulation of DJL (relying on a piecewise constant Q-function) work in sequential settings, when assigned treatments affect the future expected outcome?
Is the observed behavior assumed to be derived by a single policy? Is it known? My guess toward the second question, based on the use of propensity scores, is that it is unknown and estimated.
Is the observed outcome discrete or continuous?

The use of DJL in the toy example in Section 3.1 isn’t informative as fairly few details have been shared about the approach by that point. It’s inclusion with limited vague details was slightly confusing. The discussion in lines 136-141 helps but I think it would have been better to introduce the technical concepts first.

When mentioning the two model assumptions in Sec. 3.2 what are some specific scenarios that each assumption may cover? Being more direct at this stage may help solidify the expressed limitations of the kernel-based assumption of a smooth Q function and expose the need for DJL. Dynamic pricing is mentioned in passing earlier in the paper in regards to piecewise functions. Would medication dosing be the implied application for the second model assumption? Also, are the functions $q_{\mathcal{I}, 0}$ in model assumption 1 constant? In line 154, what does it mean for the Q(x,a) to be approximately constant?

I really appreciated the efforts by the authors to make the DJL algorithm easier to follow by breaking out the specific steps. I however found several points that were unclear. I’ve tried to highlight where I was confused in each step.

**data splitting** Are the $\mathbb{L}_\ell$ subsets disjoint? If not, how are they sampled to ensure balance? What does it mean that the different estimates will be aggregated? Is this aggregation just summing over the value estimate in Eqt. 10 and then averaging by the number of samples?  More precision here is particularly necessary as this is where the technical understanding of $\hat{D}$ is established.

**deep discretization** This step seems to be the most critical component of the algorithm. However, it is possibly the hardest to understand based on the current state of the writing. It took me several passes through the section spanning lines 201-221 to have some basic understanding of what is expected to happen. For example, it’s not clear what the change points list $\tau(v^*)$ corresponds to? Given the definition on line 211, it seems that it may be a telescoping set of indices. The recursive relationship could be described much better. Additionally, it would be nice if there was a more careful description of what $v^1$ minimizes. I assume that it corresponds to $\mathcal{C}^{(\ell)}(\mathcal{I})$ but this should be clearer. Including the pseudocode for Algorithm 1 would greatly help the presentation of this part of the paper, it should be a part of the main paper if the language in this part can’t be improved any further.

One concern that I have in terms of this step of the DJL algorithm is that it appears the deep discretization step can only reduce the size of $\hat{D}$. Meaning that there’s a potential sensitivity to the number of partitions that the interval is initialized to.

**cross-fitting** There is too little detail included with regards to how the propensity score is estimated. Additionally, the first sentence following Eqt. (10) is unclear. Is a heldout set used to estimate $\hat{V}$ that is independent from what was used to estimate the functions in the deep discretization step?

In Section 5.1, is there a typo in **S2** in the second term? I think there should be a `+` between the first indicator function and the following bracket.

On line 308, it is stated that the dimension of the features is set to 20 but in most scenarios only 2 dimensions appear to be used. Is this to add extensive noise to the estimation process? What is the purpose of this? It seems superfluous and potentially a typo?

#### **Significance**
The paper addresses relevant literature and appropriately frames its contributions within the prior work. The proposed method is well evaluated against relevant kernel-based OPE estimation approaches and is shown to have smaller bias and variance in a suite of instructive synthetic experiments as well as on a dataset derived from real-world medication dosing. The experimental and theoretical analysis of the proposed DJL show that it is an improvement over contemporary kernel-based estimation approaches.


**Time Spent Reviewing:**

15

---

> ### Author Response · Authors · 2021-08-10
> **Response to Reviewer # Ru8S (Part 2 / 2)**
>
> 5. We apologize for the confusion in the DJL algorithm. Please find our detailed responses below.
>
> 5.a). ‘Are the subsets disjoint? If not, how are they sampled to ensure balance?’
>
> Yes, the subsets are disjoint. We employ the data splitting and cross-fitting strategy in Chernozhukov et al. (2017).
>
> • *Chernozhukov, V., Chetverikov, D., Demirer, M., Duflo, E., Hansen, C. & Newey, W. (2017), 377 ‘Double/debiased/neyman machine learning of treatment effects’, American Economic Review 378 107(5), 261–65.*
>
>  5.b). ‘What does it mean that the different estimates will be aggregated? Is this aggregation just summing over the value estimate in Eqt. 10 and then averaging by the number of samples?’
>
> Aggregation is summarized in equation (10). Specifically, we evaluate the target policy in each subsample $\mathbb{L}_{\ell}$, based on the estimators (including the estimated conditional mean function $\widehat{q}_\mathcal{I}$, the estimated propensity score function $\widehat{b}_\mathcal{I}$, and the discretization $\widehat{\mathcal{D}}$) trained in its complementary subsamples $\mathbb{L}_\ell^c = \{1,\cdots,n\} - \mathbb{L}_\ell $. Denote this value estimator for subset $\mathbb{L}_\ell$ as $\widehat{V}_\ell$. The final proposed value estimator is to aggregate over $\widehat{V}_\ell$ for $\ell =1 ,\cdots,\mathcal{L}$. That’s how we aggregate value estimated over different subsets to get full efficiency. We will make this clearer shall the paper be accepted.
>
> 5.c). ‘It’s not clear what the change points list corresponds to? Additionally, it would be nice if there was a more careful description of what $\nu^1$ minimizes.’
>
> The change points list $\tau(\nu^*)$ is the set of change-point locations in $[0, \nu^*/m]$ computed by the dynamic programming algorithm. It is computed iteratively based on the update $\tau(\nu^*)=\{\nu^1, \tau(\nu^1)\}$, which means that during each iteration, it includes the current best change point location $\nu^1$ (**that minimizes equation (8)**) and the previous change-point list for the interval $[0,\nu^1/m]$. By iterating $\nu^*$ from 1 to $m$, we can find the best change-point set for interval $[0,1]$.
>
> We will include the pseudocode for Algorithm 1 in the appendix and add more descriptions to clarify our algorithm.
>
> 5.d). ‘One concern that I have in terms of this step of the DJL algorithm is that it appears the deep discretization step can only reduce the size of $\widehat{\mathcal{D}}$. Meaning that there’s a potential sensitivity to the number of partitions that the interval is initialized to.’
>
> This is an excellent comment! Generally speaking, a large $m$ leads to a small bias of value, at the cost of an increased computational time, as discussed in lines 314-318 of the main text. This tradeoff is depicted in Figure 1 in Appendix B of the supplementary article. In our paper, we require the integer $m$ to diverge with the sample size $n$, such that the conditional mean function $Q$ can be well-approximated by a piecewise function on these intervals (also see lines 160-162). In practice, we recommend setting $m$ to be proportional to the sample size $n$, i.e., $m = n/c$ for some constant $c > 0$ (see lines 165-167). The accuracy of the resulting value estimator is not overly sensitive to the choice of $c$, as long as $c$ is not too large. This is illustrated in Figure 1 in Appendix B of the supplementary article.
>
> 5.e). ‘There is too little detail included with regards to how the propensity score is estimated.’
>
> We apologize for this. In practice, any non-parametric or machine learning methods can be used to estimate the propensity score. In our simulation studies and real data application, we estimate the propensity score function use multilayer perceptron (MLP). More details are given in the source code (in *DJL.py* lines 71-79) provided in the supplementary material.
>
> 5.f). ‘Is a heldout set used to estimate $\widehat{V}$ that is independent from what was used to estimate the functions in the deep discretization step?’
>
> Yes, for each subset constructed using sample splitting, we estimate the value based on the subjects from this subset, with the estimated functions (including the estimated conditional mean function $\widehat{q}_\mathcal{I}$, the estimated propensity score function $\widehat{b}_\mathcal{I}$, and the discretization $\widehat{\mathcal{D}}$) trained from the complementary set of this subset.
>
> 6. ‘In Section 5.1, is there a typo in S2 in the second term?’
>
> Yes, thanks for pointing it out. There should be a ‘+’ between the second indicator function and the following bracket. We will correct this shall the paper be accepted.
>
> 7. ‘On line 308, it is stated that the dimension of the features is set to 20 but in most scenarios only 2 dimensions appear to be used. Is this to add extensive noise to the estimation process? What is the purpose of this? It seems superfluous and potentially a typo?’
>
> This is not a typo. The dimension of the features is set to be 20 and only 2 features are related to the conditional mean function. This is consistent with the findings in practice. In many real examples, there may exist a large number of features. However, only a few of them may be related to the outcome of interest. As you commented, those useless features are included to add extensive noise to investigate the performance of our proposed method in high-dimensional settings.
>
> **Response to Limitations Section**
>
> 1. Computational complexity required to implement the proposed approach.
>
> We analyze the computational complexity for the proposed method as follows. There are three main dominating parts of the computation: the adaptive discretization, the estimations of conditional mean function and the propensity score function, and the construction of the final value estimator.
>
> First, for the adaptive discretization on the treatment space (the main part of DJL, see Algorithm 1 Part III.4), we use the pruned exact linear time (PELT) method in Killick et al. (2012) to solve the dynamic programming (see lines 206-208). This step requires at least $\mathcal{O}(m)$ computing steps and at most $\mathcal{O}(m^2)$ steps (Friedrich et al. 2008). According to Theorem 3.2 in Killick et al. (2012), the expected computational cost is $\mathcal{O}(m)$.
>
> • *Killick, Rebecca, Paul Fearnhead, and Idris A. Eckley. "Optimal detection of changepoints with a linear computational cost." Journal of the American Statistical Association 107.500 (2012): 1590-1598.*
>
> Second, for each step in the linear complexity of adaptive discretization, we need to train the deep neural network for the conditional mean function and the propensity score function to calculate the cost function. Here, the time and space complexity of training a deep learning model vary depending on the actual architecture used. In our implementation, we employ the commonly used multilayer perceptron (MLP) to estimate the function $Q$ and the propensity score in each segment. Suppose we use the standard fully connected MLPs of $w$ width and $d$ depth with feedforward pass and back-propagation under total $e$ epochs. Then according to the complexity analysis of neural networks (see the related link below), the computational complexity of modeling the function $Q$ and the propensity score is $\mathcal{O}(2 *n e (d-1) w^2 )$.
>
> • *https://ai.stackexchange.com/questions/5728/what-is-the-time-complexity-for-training-a-neural-network-using-back-propagation/5730*
>
> For the last part, the construction of the final value estimator based on $\mathcal{L}$-fold cross fitting, which repeats the above two steps $\mathcal{L}$ times. Therefore, by putting the above results together, the total expected computational complexity of the proposed DJL is $\mathcal{O}(\mathcal{L} *m *2 *n  e (d-1)w^2)$.
>
> Note that the computation for the last part (i.e., cross-fitting) can be easily implemented in parallel computing, and thus the total expected computational complexity of the proposed DJL can be reduced to $\mathcal{O}( m *2 * n  e (d-1)w^2)$.
>
> We will include the above discussion on the computational complexity for the proposed DJL shall our paper be accepted.
>
> 2. ‘There is no clear discussion surrounding how one would be able to determine what modeling assumption to use when selecting between a piecewise of continuous Q function approximation.'
>
> We would like to clarify that there is no need to determine what modeling assumption to use in practice. These model assumptions are used to derive the theoretical properties of our method. Under different model assumptions, the rate of convergence is different. The method is essentially the same regardless of what model assumption is being imposed. It would yield a consistent value estimator, as long as either model assumption is satisfied. Please also see related discussions in the above 4.c).
>
> We will include all the above discussions and explanations shall the paper be accepted.
>
> We once again appreciate your great effort in reviewing our paper. We hope that the above discussions have addressed all your comments.

---

> > ### Comment · Reviewer_Ru8S · 2021-08-10
> > **Thank you**
> >
> > I just wanted to briefly thank the authors for addressing the various questions and suggestions included in the review I submitted of this paper. I greatly appreciate the time and attention spent to help me better understand the work prior to the discussion phase with the other reviewers.

---

> > > ### Author Response · Authors · 2021-09-02
> > > **Thank You**
> > >
> > > We want to thank you again for your thoughtful comments and suggestions! We were wondering whether our response has sufficiently addressed your concerns. Please, let us know if you have any other comments or questions. We would be happy to do our utmost to address them!

---

> > > > ### Comment · Reviewer_Ru8S · 2021-09-02
> > > > **Concerns largely addressed**
> > > >
> > > > Apologies for the drawn out waiting game that has been this reviewing process.
> > > >
> > > > I wanted to quickly relate to the authors that the major concerns and questions about clarification have been addressed by their detailed response. I appreciate the time they took with all of the reviews they received and I sincerely believe that all of the promised clarifications and additions (if included in the final paper) will warrant that the paper be published.
> > > >
> > > > One lingering concern that may be an element of future work or may even be off-base is the consideration that the interval partitioning isn't fully adaptive. What I think of when that term is used is that the change point selection may work in both directions of reducing the number of intervals as well as expanding the number of intervals as necessary. My initial stopping issue was the consideration that the reduction of the interval partitions may overfit and be unable to accurately account for some regions of the Action-Value function. I presume that this is part of the motivation for splitting the data in to disjoint subsets as well as the cross-fitting step, is to avoid this type of overfitting as well as attempt to guide some form of generalization across the various subsets of data. I appreciate the attention the authors paid to this concern and the extensive clarification to how to properly initialize the number of intervals (and empirical evidence presented in both the paper and it's supplement). One final recommendation would be for the authors to clearly reference these experiments and the discussion/clarification in the Supplement.
> > > >
> > > > All in all, I have decided to raise my rating of this paper to fully reflect my estimation that it should be accepted to the conference proceedings.

---

> ### Author Response · Authors · 2021-08-10
> **Response to Reviewer # Ru8S (Part 1 / 2)**
>
> We sincerely appreciate the reviewer's great efforts in reviewing our paper and thank the reviewer's valuable comments, many of which will lead to a more readable and self-contained version of our paper. We attempt to address all the points one by one in the following. We will revise the paper by taking into accounts all the reviewers' comments/suggestions.
>
> **Response to Quality/Clarity Section**
>
> 1. Thank you for the suggestion on improving the introduction. We summarized the limitations of kernel-based methods in lines 32-37. We will follow your suggestion to revise the introduction shall our paper be accepted. For instance, we will summarize the findings in our toy example (see Section 3.1 of the main text) to specify the limitations of kernel-based methods. The proposed DJL method adaptively discretizes the action space. The discretization addresses the first limitation of kernel-based methods, allowing us to handle discontinuous value functions. The adaptivity addresses the second limitation of kernel-based methods. Specifically, it guarantees the optimality of the proposed method in cases where the second-order derivative of the conditional mean function has an abrupt change in the treatment space.
>
> 2. Response to questions in Section 2.1.
>
> 2.a). ‘Are the feature-treatment-outcome observations fixed in time? In other words, is the problem set up to only consider single step treatment scenarios?’
>
> Yes, we consider policy evaluation in a single time step, where feature-treatment-outcome observations are fixed in time and independent across different subjects, also known as batched contextual bandits setting. This setting is in accordance with the cited OPE works in lines 22-26.
>
> 2.b). ‘As a follow-up, what are the limitations of DJL when considering the extension to sequential decision making? Does the formulation of DJL (relying on a piecewise constant Q-function) work in sequential settings, when assigned treatments affect the future expected outcome?’
>
> First, the proposed method can be extended to sequential decision-making with continuous action space. Specifically, our procedure can be coupled with the fitted Q-iteration algorithm (Riedmiller, 2005) that iteratively estimated the Q-function using deep jump learning. However, a potential limitation is that it would be more computationally intensive, as fitted Q-iteration requires iteratively updating the Q-function. It remains unclear how to facilitate the computation in sequential decision-making. We leave this for future investigation.
>
> • *Riedmiller, Martin. "Neural fitted Q iteration–first experiences with a data efficient neural reinforcement learning method." European conference on machine learning. Springer, Berlin, Heidelberg, 2005.*
>
> 2.c). ‘Is the observed behavior assumed to be derived by a single policy? Is it known? My guess toward the second question, based on the use of propensity scores, is that it is unknown and estimated.’
>
> Yes, the behavior/logging policy that generates the observed data is single BUT can be unknown and estimated from the data. It is possible to extend our work to settings with multiple logging policies from stratified sampling. Please see a recent relevant work below.
>
> • *Kallus, Nathan, Yuta Saito, and Masatoshi Uehara. "Optimal off-policy evaluation from multiple logging policies." International Conference on Machine Learning. PMLR, 2021.*
>
> 2.d). 'Is the observed outcome discrete or continuous?'
>
> The observed outcome can be either discrete or continuous. We did not impose additional assumptions.
>
> 3. Thank you for your suggestion on improving the organization of Section 3.1. The main purpose of the toy example is to visualize the limitation of the kernel-based method and demonstrate our motivation. We will include more technical concepts, such as the definition of discretization and piecewise approximation of the conditional mean function $Q$ from Section 3.2 to improve the readability of this section.
>
> 4. Thank you for the suggestion on improving Section 3.2!
>
> 4.a). ‘When mentioning the two model assumptions in Sec. 3.2 what are some specific scenarios that each assumption may cover?’
>
> Model 1 (piecewise function) covers the dynamic pricing example we mentioned in the introduction (see also lines 243-244). In Scenarios 1 and 2 of our simulation studies (see also line 305), the underlying model is set to be a piecewise function. Model 2 (continuous function) covers the personalized dose-finding example. In Scenarios 3 and 4 of our simulation studies (see also lines 305-306) as well as the real data section, the underlying model is set to be a continuous function. We will emphasize these examples when we first impose two model assumptions to solidify the limitations of the kernel-based method and motivate the proposed DJL.
>
> 4.b). ‘Would medication dosing be the implied application for the second model assumption?’
>
> Suppose the treatment effect would vary smoothly as the dose changes, then the dosing example would satisfy the second model assumption.
>
> 4.c). ‘Also, are the functions in model assumption 1 constant? In line 154, what does it mean for the Q(x,a) to be approximately constant?’
>
> Under Model 1, we assume the function $Q(x,a)$ is a piecewise function on the action space. Within each segment $\mathcal{I}$, the function $Q(x,a)$ is a constant function of $a$, but can be any function of the features $x$. In other words, $Q(x,a_1) = Q(x,a_2)$ for any $a_1,a_2 \in \mathcal{I}$. Thus, we denote the function $Q(x,a)$ at each segment $\mathcal{I}$ as $q_\mathcal{I}(x)$, which yields a piecewise function $Q(x,a) = \sum_{\mathcal{I}} q_\mathcal{I}(x) \mathbb{I}(a \in \mathcal{I})$, as stated in equation (4) in lines 149 -150. In the real applications, the true function $Q(x,a)$ could be either a continuous function, or a piecewise function.  As such, we propose to approximate the underlying unknown function $Q(x,a)$ by these piecewise functions of $a$ using the proposed DJL. It works when either Model 1 or 2 holds. We further show theoretical guarantee of such approximation. Please refer to our Theorems 1 and 2 for details. We will include these elaborations shall our paper be accepted.

---

### Official Review · Reviewer_54Kf · 2021-07-15

**Rating:** 6
**Confidence:** 3

**Summary:**

The paper studies off-policy evaluation (OPE) with continuous actions (not the prevalent discrete action settings). To handle continuous treatments, the paper develops a new estimation method for OPE using deep  jump learning. The key idea is to adaptively discretize the continuous treatment space based on deep learning and multi scale change point detection. The proposed method is compared with some previous related methods from both theoretical and empirical perspectives.

**Limitations And Societal Impact:**

One concern is that the paper does not touch on some recent related works such as [a] and [b]. I think that the paper still makes solid contributions given these additional related works. Thus, it would be good for the authors to just cite these papers and clarify the difference.

[a] Nathan Kallus and Masatoshi Uehara. Doubly Robust Off-Policy Value and Gradient Estimation for Deterministic Policies. In Advances in Neural Information Processing Systems, 33 (NeurIPS 2020).

[b] Mert Demirer, Vasilis Syrgkanis, Greg Lewis, and Victor Chernozhukov. Semi-Parametric Efficient Policy Learning with Continuous Actions. In Advances in Neural Information Processing Systems, 32 (NeurIPS 2019).


Another point that I would like to state here is that the simulation study is extremely simple. Specifically, the methods are tested only on small sample settings (i.e., n=50-300). Thus, there remains some concerns about the empirical performance of the proposed procedure. For example, the authors suggest that $m=n/10$ to achieve a good balance between the bias and the computational cost. This seems to work with n=50-300, but it is unclear that the same suggestion is applied to larger samples such as n=1000,10000,.... Showing how the method works in large sample settings is also critical as it is general in the personalized pricing problem in the web industry. The paper also states that a longer computational time (compared to Colangelo & Lee) might be a potential limitation. Arguing this potential limitation is nice, however, it is desirable to test how many samples DJL can handle to better see its applicability.

**Main Review:**

The paper studies the important problem of OPE with continuous actions, which is also of interest to the NeurIPS audience. The paper nicely argues the weakness of the previous kernel-based approaches in Section 3.1 “Toy Example”. This toy example well motivates the proposed approach. Then, a novel adaptive discretization using deep learning is proposed. One advantage of DJL is that it does not need the tuning of bandwidth. The proposed DJL is theoretically analyzed and shown to converge at a faster rate compared to kernel-based estimators. Overall, the paper is well-written and the motivation is easy to follow.

**Time Spent Reviewing:**

6

---

> ### Author Response · Authors · 2021-08-10
> **Response to Reviewer # 54Kf**
>
> We greatly appreciate the reviewer's effort and valuable comments, many of which will lead to a more readable and self-contained version of our paper. We attempt to address all the points one by one in the following. We will revise the paper by taking into accounts all the reviewers' comments/suggestions.
>
> **Recent Related Works**
>
> Thank you for pointing out two recent related papers. We will cite these papers and clarify the difference shall our paper be accepted.
>
> First, the paper [a] (Kallus and Uehara, 2020) considered off-policy evaluation and gradient estimation in reinforcement learning. It proposes to use the kernel method to handle continuous actions. In our paper, we consider a contextual bandit setting and do not use the kernel method. It would be practically interesting to extend our adaptive discretization method to the reinforcement learning setting. We leave it for future research. Second, the paper [b] (Demirer, et al., 2019) proposed a semi-parametric approach for offline policy evaluation and optimization in the contextual bandit setting. To handle the continuous action space, they imposed a parametric assumption on the value function. In our paper, we did not impose such an assumption. Instead, we propose to approximate the value by a piecewise constant function of the action space.
>
> **Simulation Study Settings**
>
> Thanks for the comment. We would like to clarify that we focus on the setting with $n=50-300$ to be consistent with the experimental setting used in Kallus & Zhou (2018), which is one of the competing methods considered in our paper. Please refer to Section 5 of Kallus & Zhou (2018) for more details. We note that the proposed method performs well even when the sample size is small ($n = 50$), while the kernel-based methods would fail to accurately estimate the value even when $n = 300$.
>
> • *Kallus, N. & Zhou, A. (2018), Policy evaluation and optimization with continuous treatments, in ‘International Conference on Artificial Intelligence and Statistics’, PMLR, pp. 1243–1251.*
>
> **Choice of $m$ under Different Sample Sizes**
>
> We acknowledge that the number of initial intervals $m$ represents a trade-off between the estimation bias and the computational cost. We discussed this tradeoff in lines 314-318 of the main text. In practice, we recommend to set $m = n/10$. When $n$ is small, the performance of the resulting value estimator is not overly sensitive to the choice of $c$ as long as $c$ is not too large. Please see the left panel of Figure 1 in Appendix B of the supplementary article for details. When $n$ is large, we further investigate the computational capacity by setting $m=n/10$ for large sample sizes $n=1000-10000$. It turns out that such a choice of $c$ can still handle datasets with a few thousand observations. Please see our response to your next comment. Finally, when $n$ is extremely large, setting $m=n/10$ might be computationally intensive. In that case, we can adopt some other methods (e.g., intelligence sampling) to further facilitate the computation. Please refer to our response to your next comment for details.
>
> **Computational Time under Large Sample Size**
>
> We agree large sample settings may occur in the practice. In the following, we implement our method to settings where $n=1000-10000$ and report the corresponding computational time. The computing infrastructure was stated in lines 294-295 of the main text. We use Scenario 1 and consider the sample size chosen from $n \in [1000, 2000, 5000, 10000]$ for illustration.
>
> |   Sample Size   |      $n=1000$      |   $n=2000$    |        $n=5000$      |      $n=10000$ |
> |----------|:-------------:|------:|------:|------:|
> |  **Computational Time**       |     15.92 Minutes   |    30.40 Minutes   |    1.32 Hours        |     2.86 Hours |
>
>
> Here, we use parallel computing to process each fold, as our algorithm employs data splitting and cross-fitting. This largely facilitates the computation, leading to shorter computation time compared to those listed in Table 1 in Appendix B.
>
> In addition to parallel computing, there are some other techniques we can use to handle datasets with large sample sizes. For instance, in the changepoint literature, Lu (et al., 2017) proposed an intelligence sampling method to identify multiple change points with long time series data. Their method would not lose much statistical efficiency, but is much more computationally efficient. It is possible to adopt such an intelligence sampling method to our setting for adaptive discretization. This would enable our method to handle large datasets. We will leave it for future investigation.
>
> • *Lu, Zhiyuan, Moulinath Banerjee, and George Michailidis. "Intelligent sampling for multiple change-points in exceedingly long time series with rate guarantees." arXiv preprint arXiv:1710.07420 (2017).*
>
> We will include all the above discussions shall our paper be accepted.
>
> We once again appreciate your effort in reviewing our paper. We hope that the above discussion has addressed your comments.

---

> > ### Author Response · Authors · 2021-09-02
> > **Thank You**
> >
> > We want to thank you again for your thoughtful comments and suggestions! We were wondering whether our response has sufficiently addressed your concerns. Please, let us know if you have any other comments or questions. We would be happy to do our utmost to address them!

---

### Official Review · Reviewer_cawi · 2021-07-16

**Rating:** 8
**Confidence:** 4

**Summary:**

The paper considered an off-policy evaluation method in continuous treatment settings and corresponding asymptotic properties are established. The experimental results are exciting.

**Limitations And Societal Impact:**

Yes

**Main Review:**

Overall, the paper is well-written, and is very potential to be published in NeurIPS. However, some mathematical expressions, while correct, are hard to interpret, e.g. equation (8) and (9). It would be nice if the authors could provide a brief explanation in words.

**Time Spent Reviewing:**

5

---

> ### Author Response · Authors · 2021-08-10
> **Response to Reviewer # cawi**
>
> We greatly thank the reviewer's effort and positive comments on our paper. Please find our response to your valuable comments in the following. We will revise the paper by taking into accounts all the reviewers' comments/suggestions.
>
> **Mathematical Expressions**
>
> We apologize for the ambiguous interpretation of the mathematical expressions. Below, we make a few clarifications. First, we recursively define the Bellman cost function in equation (8), for $\nu^*=1,2,\cdots$. For a given $\nu$, the right-hand side corresponds to the cost of partitioning on a particular point. We then identify the best $\nu$ that minimizes the cost. This yields the Bellman function on $[0, \nu^*/m]$ on the left-hand side. In other words, equation (8) is a recursive formula used in our dynamic algorithm to update the Bellman equation for the locations of change points. By ‘recursive’, notice that the Bellman function appears on both sides of equation (8).
>
> Equation (9) restricts the research space in (8) to a potentially much smaller set of candidate change points, i.e., $\mathcal{R}_{v*}$. The main purpose is to facilitate the computation. It yields a linear computational cost (Killick et al. 2012). In contrast, without these restrictions, it would yield a quadratic computational cost (Friedrich et al. 2008).
>
> We will include the related discussions shall the paper be accepted. We once again appreciate your effort in reviewing our paper. We hope that the above discussion has addressed your comments.

---

> > ### Author Response · Authors · 2021-09-02
> > **Thank You**
> >
> > We want to thank you again for your thoughtful comments and suggestions! We were wondering whether our response has sufficiently addressed your concerns. Please, let us know if you have any other comments or questions. We would be happy to do our utmost to address them!

---

### Official Review · Reviewer_P63y · 2021-07-17

**Rating:** 6
**Confidence:** 3

**Summary:**

The authors propose a new approach to off-policy evaluation for contextual bandits with continuous action space based on adaptive discretization of the action space. The authors mainly compare the proposed approach with approaches based on fixed-bandwidth kernel smoothing of the action space and show both theoretically and empirically that the proposed approach performs better in certain classes of value functions.

**Limitations And Societal Impact:**

Yes


**Main Review:**

While the differences between piecewise constant approximation and a smooth kernel are well-known, the use of piecewise constant action-value estimation in this particular setting seems new to me. I believe the contribution is significant enough for publication in NeurIPS.

For fixed-bandwidth kernels, there is only one parameter to estimate whereas the proposed approach needs to evaluate a potentially large number of action-space partitions. Is this the main price to pay for the proposed approach?

What about adaptive kernels with multiple bandwidths? Also, what about a tree-based discretization as in [Majzoubi et al 2020 Efficient Contextual Bandits with Continuous Actions] ?

Perhaps the authors can elaborate on the above.

**Time Spent Reviewing:**

2

---

> ### Author Response · Authors · 2021-08-10
> **Response to Reviewer # P63y**
>
> We greatly appreciate the reviewer's effort and valuable comments, many of which will lead to a more readable and self-contained version of our paper. We attempt to address all the points one by one in the following. We will revise the paper by taking into accounts all the reviewers' comments/suggestions.
>
> **Main Price of the Proposed Approach**
>
> Yes, the main price to pay for the proposed approach is the relatively long computational time to find the action-space partitions, compared with directly setting one single bandwidth. This potential limitation is included in the paper as discussed in lines 335-338. Please refer to Table 1 in the supplementary article for the detailed computational cost for each of the methods.
>
> **Potential Alternative Approaches**
>
> This is an excellent comment! Thank you for introducing two possible alternative approaches. First, our proposed method can be understood as an adaptive kernel method with the boxcar kernel function, since we adaptively discretize the action space to compute a set of intervals with different lengths. It would be practically interesting to extend our proposal to consider more general kernel functions. However, this is beyond the scope of the current paper and we leave it for future research. Second, the paper by Majzoubi et al (2020 Efficient Contextual Bandits with Continuous Actions) mainly focused on online and offline policy optimization in contextual bandits and proposed a tree-based discretization to handle continuous actions. Extending the tree-based discretization with adaptive pruning in offline policy evaluation is another possible direction to handle our problem. We will include these discussions shall our paper be accepted.
>
> We once again appreciate your effort in reviewing our paper. We hope that the above discussion has addressed your comments.

---

> > ### Author Response · Authors · 2021-09-02
> > **Thank You**
> >
> > We want to thank you again for your thoughtful comments and suggestions! We were wondering whether our response has sufficiently addressed your concerns. Please, let us know if you have any other comments or questions. We would be happy to do our utmost to address them!

---

### Decision · Program_Chairs · 2021-09-28

**Decision:**

Accept (Poster)

**Comment:**

This paper presents an off-policy evaluation method motivated by medical treatment settings, and provides a theoretical analysis as well as simulation experiments.  The reviewers support the paper overall and I recommend acceptance; that said, many refined points were raised during discussion, and I highlight the reviews (and discussion) by reviewers Ru8S and 54Kf.  The authors responded to these concerns extensively below, but I request they incorporate these discussions and suggestions in their revisions.

**Consistency Experiment:**

NeurIPS has a long history of experimentation. In 2014, NeurIPS ran an experiment in which 10% of submissions were reviewed by two independent committees to quantify the randomness in the review process. This year, we repeated a variant of this experiment to see how the quality of the review process has changed over time.  This paper was part of the experiment and was therefore assigned to two committees (consisting of reviewers, an Area Chair, and a Senior Area Chair) that reached independent decisions.  If both committees made the same recommendation, this recommendation was followed. If a single committee recommended acceptance, the paper was accepted (with the exception of a few cases in which the other committee identified what we considered a fatal flaw, e.g., an error in a key result).

Both committees reached the same decision: **Accept (Poster)**

The other committee assigned to the paper recommended **Accept (Poster)**.  You can find the other set of reviews, along with any follow up discussion with the authors here:
https://openreview.net/forum?id=rvKD3iqtBdk